# TonEBP/NFAT5 promotes obesity and insulin resistance by epigenetic suppression of white adipose tissue beiging

Hwan Hee Lee[1], Seung Min An[1], Byeong Jin Ye[1], Jun Ho Lee[1], Eun Jin Yoo[1], Gyu Won Jeong[1], Hyun Je Kang[1], Assim A. Alfadda[2], Sun Woo Lim[3], Jiyoung Park [1], Whaseon Lee-Kwon[1], Jae Bum Kim [4], Soo Youn Choi [1] & Hyug Moo Kwon[1]

Tonicity-responsive enhancer binding protein (TonEBP or NFAT5) is a regulator of cellular adaptation to hypertonicity, macrophage activation and T-cell development. Here we report that TonEBP is an epigenetic regulator of thermogenesis and obesity. In mouse subcutaneous adipocytes, TonEBP expression increases > 50-fold in response to high-fat diet (HFD) feeding. Mice with TonEBP haplo-deficiency or adipocyte-specific TonEBP deficiency are resistant to HFD-induced obesity and metabolic defects (hyperglycemia, hyperlipidemia, and hyper-insulinemia). They also display increased oxygen consumption, resistance to hypothermia, and beiging of subcutaneous fat tissues. TonEBP suppresses the promoter of β3-adrenoreceptor gene, a critical regulator of lipolysis and thermogenesis, in ex vivo and cultured adipocytes. This involves recruitment of DNMT1 DNA methylase and methylation of the promoter. In human subcutaneous adipocytes TonEBP expression displays a correlation with body mass index but an inverse correlation with β3-adrenoreceptor expression. Thus, TonEBP is an attractive therapeutic target for obesity, insulin resistance, and hyperlipidemia.

[1] School of Life Sciences, Ulsan National Institute of Science and Technology, Ulsan 44919, Republic of Korea. [2] Obesity Research Center, College of Medicine, King Saud University, Riyadh 11461, Saudi Arabia. [3] Transplantation Research Center, Catholic University of Korea, Seoul 03083, Republic of Korea. [4] National Creative Research Initiatives Center for Adipose Tissue Remodeling, Institute of Molecular Biology and Genetics, Department of Biological Sciences, Seoul National University, Seoul 08826, Republic of Korea. Correspondence and requests for materials should be addressed to S.Y.C. (email: sychoi@unist.ac.kr) or to H.M.K. (email: hmkwon@unist.ac.kr)

Energy balance is an important factor in the pathogenesis of metabolic disease. If energy intake exceeds energy expenditure, weight gain and ultimately obesity result. Excess energy is stored as lipids in adipose tissue, but unfettered expansion of adipose tissue can result in a pathological condition characterized by local hypoxia, which leads to insulin resistance, impaired thermogenic activity, and chronic inflammation[1,2]. Adipose tissue is also capable of transforming chemical energy into heat, thereby opposing excessive fat accumulation, through the activity of specialized thermogenic adipocytes. The classical brown adipocytes in brown adipose tissue constitutively express thermogenic genes. In addition, brown-like adipocytes or beige cells are also present in white adipose tissue. Thermogenesis can be induced in these cells in response to a variety of activators[3]. Activation of the thermogenic gene program in brown or beige adipocytes increases systemic energy expenditure and can ameliorate or prevent the development of obesity-associated metabolic disorders as a result. Thus, efforts aimed at gaining a deeper understanding of the regulation of energy storage, mobilization, and use by adipocytes may lead to the identification of therapy for metabolic disease[4].

Tonicity-responsive enhancer-binding protein (TonEBP), also known as nuclear factor of activated T cells 5 (NFAT5), belongs to the Rel family of transcriptional factors, which includes nuclear factor κB and NFAT1–4[5,6]. TonEBP was initially identified as the central regulator of cellular response to hypertonic stress[5,7,8]. Recent studies reveal that TonEBP is involved in the M1 activation of macrophages, because it promotes the expression of pro-inflammatory genes and suppresses anti-inflammatory cytokines in response to toll-like receptor-4 activation[9–11] or hyperglycemia[12]. TonEBP haplo-deficient mice show lower levels of inflammation and are resistant to inflammatory and autoimmune diseases, including sepsis[10], rheumatoid arthritis[13,14], atherosclerosis[15], encephalomyelitis[16], and diabetic nephropathy[12]. Interestingly, TonEBP suppresses peroxisome proliferator-activated receptor-2 (PPARγ2) expression during adipocyte differentiation, suggesting that TonEBP is involved in controlling adipogenesis[17,18].

MicroRNAs (miRs), short noncoding RNAs, block translation or induce degradation of the target mRNAs by base-pairing to complementary sequences[19]. Recently, the associations between TonEBP and miRs have been reported in cancer and diabetes[20–22], but there has been no report on adipose tissues. Here, we find that elevated expression of TonEBP due to downregulation of miR-30 in adipocytes leads to the development of obesity and insulin resistance, whereas its ablation enhances adipocyte beiging and prevents ectopic deposition of triglycerides.

## Results

**Adipose TonEBP expression is elevated in obesity.** TonEBP is widely expressed in many tissues. Interestingly, *TonEBP* mRNA and protein expression was dramatically escalated in inguinal white adipose tissue (iWAT), and to a lesser extent in epididymal white adipose tissue (eWAT), of mice fed with a high-fat diet (HFD; 60% energy as fat) compared with animals fed with a chow diet (CD; 10% energy as fat) (Fig. 1a, b). Likewise, *TonEBP* mRNA expression was elevated in iWAT from 10-week-old *db/db* (*Lepr^{db/db}*) obese mice compared with their lean *db/+* littermates (Supplementary Fig. 1a). In addition, the HFD-fed mice exhibited lower mRNA expression of thermogenic genes—PPARγ coactivator 1α (*PGC1-α*), *Dio2, CPT1α, Cidea, PPARα*, and β3-adrenergic receptor (*Adrb3*)—and beiging markers—*UCP-1, CD137*, and *TMEM26*—in iWAT (Supplementary Fig. 1b and c). In a group of human subjects, *TonEBP* mRNA expression in subcutaneous adipocytes correlated positively with body mass index (BMI) (Fig. 1c). Taken together, these data suggest that

TonEBP expression in iWAT is associated with obesity and suppression of thermogenic gene expression.

The microRNA-30 (miR-30) family is a potent regulator of thermogenesis in adipocytes[23], and miR-30 expression is reduced in obese mice[24]. Using prediction algorithms in TargetScan Mouse 6.2 (www.targetscan.org), we found that *TonEBP* mRNA was a potential target of miR-30 with two binding sites for miR-30b and miR-30c in its 3′UTR (Supplementary Fig. 1d). *TonEBP* mRNA and protein expression was significantly reduced in 3T3-L1 adipocytes transfected with miR-30b or miR-30c mimic (Fig. 1d, e). Consistent with this, miR-30b or miR-30c mimic enhanced mRNA expression of thermogenic genes (Supplementary Fig. 1e). To identify direct binding to TonEBP 3′UTR of miR-30b and miR-30c, we constructed luciferase vector psiCHECK-2 fused with two putative binding sites on TonEBP 3′UTR region. The luciferase activity was reduced by transfection with miR-30b or miR-30c mimic (Supplementary Fig. 1f), indicating direct binding of miR-30b and miR-30c to the 3′ UTR of TonEBP. These data demonstrate that the reduced expression of miR-30b and miR-30c in obesity contributes to the escalation of TonEBP expression and the suppression of thermogenic genes. Thus, miR-30 promotes thermogenesis and beiging via suppression of two transcriptional regulators—TonEBP and RIP140[23].

**TonEBP haplo-deficient mice are resistant to obesity.** To investigate the physiological role of TonEBP, TonEBP haplo-deficient mice (*TonEBP^{+/Δ}*), and their wild-type littermates (*TonEBP^{+/+}*) were fed with either a CD or HFD starting at 8 weeks of age. When fed with the CD, the two groups of animals showed similar weight gain (Fig. 1f). On the other hand, when fed with the HFD, the TonEBP haplo-deficient animals were resistant to weight gain (Fig. 1f, g), despite comparable food intake (Supplementary Fig. 2a). Whole-body echo MRI analysis was performed to analyze fat content. The lower body mass of TonEBP haplo-deficient mice was accounted for largely by a reduction in fat mass, without alterations in body length or lean body mass (Fig. 1h). Examination of dissected fat tissues confirmed that epididymal, inguinal, and dorsal fat pads were smaller in these animals (Fig. 1i). Likewise, TonEBP haplo-deficiency on the *Lepr^{db/db}* background was protected against weight gain and adiposity (Supplementary Fig. 2b, c). Thus, TonEBP deficiency resists obesity by interfering with fat expansion.

**Adipocyte TonEBP suppresses thermogenesis and beiging of WAT.** Reduction in adipose tissue mass without alterations in food intake (Fig. 1i and Supplementary Fig. 2a) suggested an increase in energy expenditure by the TonEBP haplo-deficient mice. Analyses of these animals by indirect calorimetry revealed higher oxygen consumption rate (VO₂) and carbon dioxide production rate (VCO₂) (Fig. 2a, b), with no changes in respiratory exchange ratio (RER) (Supplementary Fig. 3a). Consistent with the higher VO₂, these animals generated more heat (Fig. 2c) and had an ~0.6 °C higher body temperature (Fig. 2d). In addition, the body temperature remained higher when exposed to cold (4 °C) in CD- (Fig. 2e) and HFD-fed mice (Supplementary Fig. 3b), suggesting more efficient adaptive thermogenesis.

Reduced thermogenesis is a characteristic feature of obesity in humans and mice[25]. We asked whether TonEBP affected thermogenesis and browning of adipocytes. The TonEBP haplo-deficient mice had higher expression of thermogenic genes (Fig. 2f), and beige marker genes in iWAT (Fig. 2g), but not in BAT (Supplementary Fig. 3c). This pattern of higher expression was largely maintained after the animals were exposed to cold (Fig. 2h and Supplementary Fig. 3d). Histological and

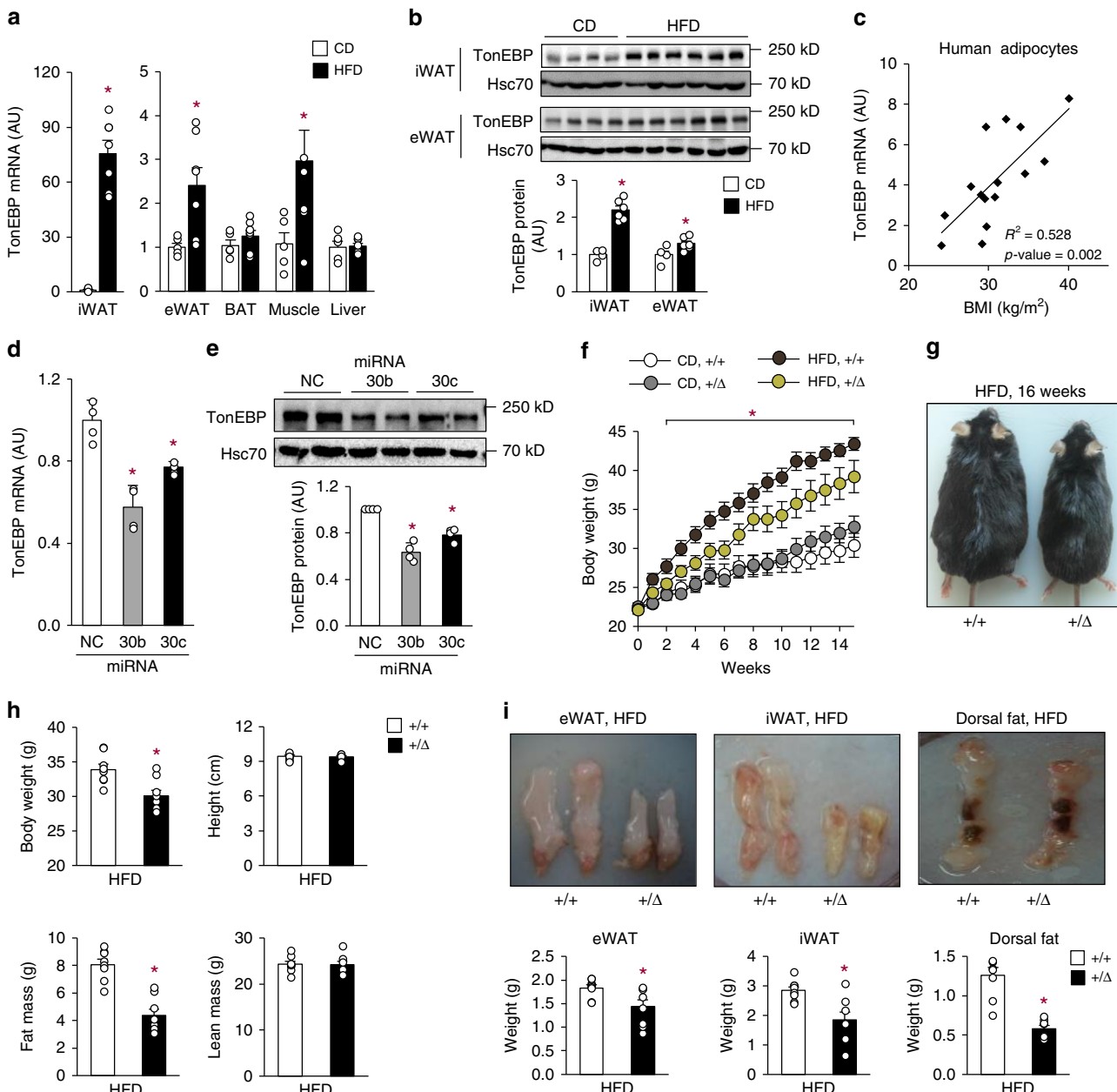

**Fig. 1** Adipocyte TonEBP expression is elevated in obesity and TonEBP-deficient mice resist obesity. **a** *TonEBP* mRNA levels in iWAT, eWAT, BAT, muscle, and liver from C57BL/6 J mice fed with CD (chow diet, $n = 5$) or HFD (high-fat diet, $n = 7$) for 16 weeks. **b** Immunoblots (**top**) and quantification of protein levels (**bottom**) of TonEBP and Hsc70 in iWAT and eWAT (CD, $n = 4$; HFD, $n = 6$). **c** Correlation of *TONEBP* mRNA levels in human subcutaneous adipocytes and BMI ($n = 15$). *TonEBP* mRNA (**d**) and representative immunoblots (**e**, **top**) and quantification of protein levels (**e**, **bottom**) in 3T3-L1 adipocytes transfected with miR-negative control (NC), miR-30b (30b), or miR-30c (30c) ($n = 4$). **f–i** *TonEBP*$^{+/\Delta}$ mice ( $+/\Delta$) and their *TonEBP*$^{+/+}$ littermates ($+/+$) were fed with CD or HFD as indicated. **f** Changes in body weight after a switch to HFD (CD+/+, $n = 7$; CD $+/\Delta$, $n = 7$; HFD+/+, $n = 13$; HFD+/$\Delta$, $n = 11$). **g** Representative images of animals fed with HFD. **h** Body weight, height, fat mass, and lean mass ($n = 8$). **i** Representative images (**top**) and weight (**bottom**) of fat pads from HFD-fed animals ($n = 4$). $n$ represents number of biologically independent samples (**a–c**, **i**) or animals (**f–h**) or independent experiments with triplicate (**d**, **e**). All data are presented as mean + s.e.m. (**a**, **b**, **f**, **h**, **i**) or + s.d. (**d**, **e**). AU, arbitrary unit. The *p*-values are determined by unpaired *t*-test (**a**,**b**, **h**, **i**) or one-way ANOVA (**d–f**). *$p < 0.05$ vs. CD (**a**), NC (**d**), or $+/+$ (**f**, **h**, **i**). Source data are provided as a Source Data file

immunohistochemical analysis of iWAT revealed enhanced features of beiging WAT in these animals—multilocular lipid droplets and higher expression of UCP-1 (Fig. 2i).

Next, we examined adipocytes in vitro. Adipocytes differentiated from stromal vascular fractions (SVF) of the TonEBP haplo-deficient animals showed elevated expression of thermogenic genes, hormone-sensitive lipase (HSL), and beige marker genes (Fig. 2j, k and Supplementary Fig. 4a). In addition, TonEBP knockdown (Supplementary Fig. 4b) enhanced, while adenovirus-mediated overexpression of TonEBP reduced the basal and isoproterenol-stimulated expression of these genes in differentiated 3T3-L1 cells (Supplementary Fig. 4c). These findings demonstrate that TonEBP suppresses energy expenditure by blocking thermogenesis and beiging of WAT.

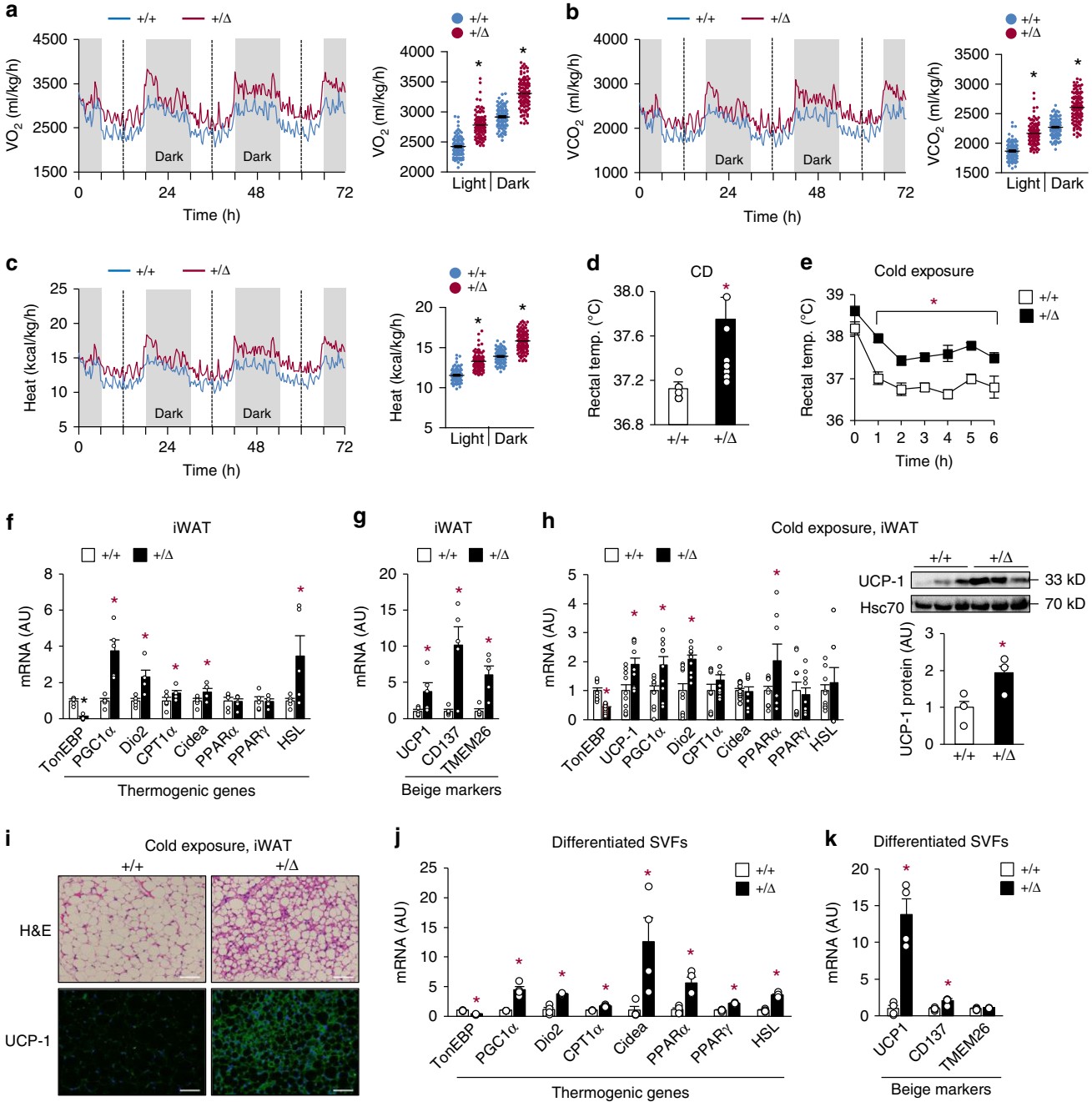

**Fig. 2** TonEBP deficiency promotes energy expenditure and beiging of WAT. HFD-fed animals were analyzed by indirect calorimetry to obtain $VO_2$ (**a**), $VCO_2$ (**b**), and heat production (**c**) ($n = 4$). Rectal temperature (temp.) measured in CD-fed animals at room temperature ($n = 8$) (**d**) and after exposure to cold up to 6 h as indicated (4 °C) ($n = 6$) (**e**). **f, g** mRNA abundance of thermogenic genes (**f**) and beiging marker genes (**g**) in iWAT of HFD-fed animals ($n = 5$). **h** mRNA abundance of thermogenic genes (left, $n = 10$) and immunoblots of UCP-1 and Hsc70 (right, $n = 3$) in iWAT of CD-fed animals exposed to cold (4 °C). **i** Representative images of iWAT sections stained with H&E and UCP-1 antibody from CD-fed animals exposed to cold (4 °C). Scale bars, 100 μm. **j, k** Thermogenic gene (**j**) and beige marker (**k**) mRNA abundance in beige adipocytes differentiated from the stromal vascular cells of iWAT ($n = 4$). $n$ represents number of biologically independent animals (**a–e**) or samples (**f–k**). **a–h, j, k** All data are presented as mean + s.e.m. AU arbitrary unit. The $p$-values are determined by unpaired $t$-test (**d, f–k**) or one-way ANOVA (**a–c, e**). *$p < 0.05$ vs. +/+. Source data are provided as a Source Data file

**TonEBP haplo-deficient mice resist obesity-induced metabolic dysfunction**. We next assessed glucose homeostasis and insulin sensitivity in the HFD-fed animals. The TonEBP haplo-deficient mice maintained lower fasting glucose levels than their WT littermates from week 1 to 16 of HFD feeding (Fig. 3a) and showed improved glucose tolerance and insulin sensitivity (Fig. 3b). After 16 weeks of HFD feeding, the TonEBP haplo-deficient mice showed the lack of increase in circulating insulin levels seen in

their WT littermates (Fig. 3c). Likewise, $Lepr^{db/db}$ obese mice with TonEBP haplo-deficiency had lower blood glucose levels than their WT littermates (Supplementary Fig. 5a). In order to assess insulin signaling directly, Akt phosphorylation in response to insulin administration was examined in the liver and eWAT, where HFD-induced inflammation causes local insulin resistance and changes in adipokine expression[26]. Phosphorylation was significantly higher in eWAT and liver of the TonEBP

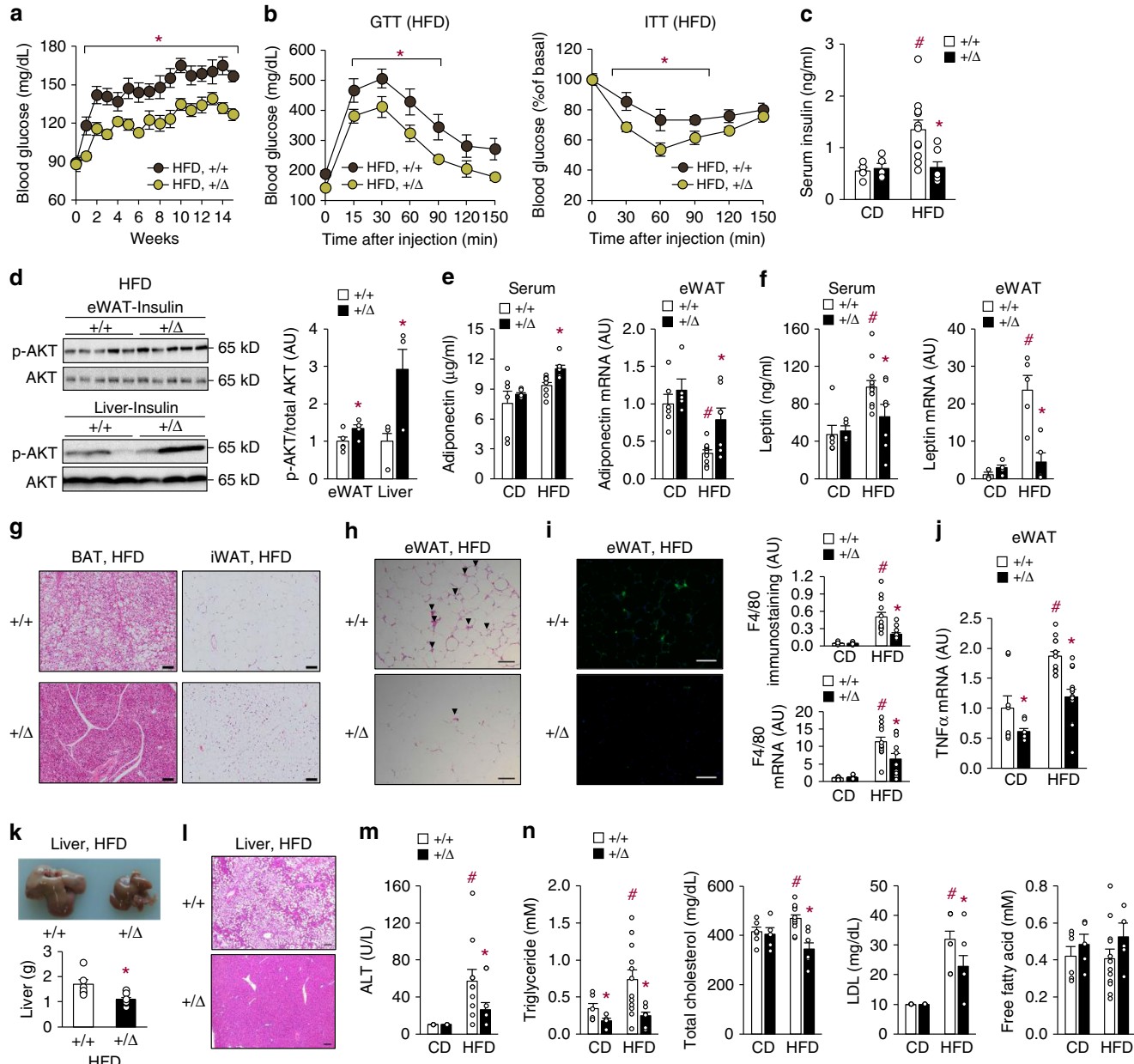

**Fig. 3** TonEBP deficiency ameliorates obesity-induced insulin resistance and metabolic dysfunction. **a** Changes in fasting blood glucose levels after a switch to HFD (+/+, $n = 13$; +/Δ, $n = 11$). **b** Glucose tolerance (left, $n = 8$) and insulin tolerance (right, $n = 9$) after 9 weeks on HFD. **c–n** Animals were analyzed after 16 weeks on HFD. **c** Circulating insulin concentrations (CD +/+, $n = 5$; CD +/Δ, $n = 5$; HFD +/+, $n = 11$; HFD +/Δ, $n = 7$). **d** Phosphorylation of AKT in eWAT ($n = 5$) and liver ($n = 3$) in response to insulin administration. **e, f** Serum concentration and mRNA abundance in eWAT of adiponectin (CD+ /+, $n = 6$; CD+ /Δ, $n = 5$; HFD+ /+, $n = 9$; HFD+ /Δ, $n = 7$) (**e**) and serum concentration (CD+ /+, $n = 6$; CD +/Δ, $n = 4$; HFD+ /+, $n = 12$; HFD+/Δ, $n = 7$) and mRNA abundance ($n = 5$) in eWAT leptin (**f**). **g** Representative images of H&E-stained sections of BAT and iWAT. **h** Representative images of H&E-stained sections of eWAT. **i** Representative images of F4/80 immunostaining (left), quantitative analysis of F4/80 immunostaining (right, top), and F4/80 mRNA levels (right, bottom) of eWAT (CD, $n = 8$; HFD, $n = 12$). **j** TnFα mRNA levels in eWAT (CD, $n = 8$; HFD, $n = 12$). **k** Representative images (top) and liver weight (bottom) (+/+, $n = 6$; +/Δ, $n = 8$). **l** Representative images of H&E-stained liver sections. ALT concentrations (CD, $n = 5$; HFD, $n = 10$) (**m**), and triglyceride, cholesterol (total and LDL), and free fatty acid levels (CD+ /+, $n = 6$; CD+/Δ, $n = 5$; HFD+/+, $n = 12$; HFD+ /Δ, $n = 7$) (**n**). $n$ represents number of biologically independent samples. All data are presented as mean + s.e.m. Scale bars, 100 μm (**g–i, l**). AU arbitrary unit. The $p$-values are determined by unpaired $t$-test (**d, k**) or one-way ANOVA (**a–c, e, f, l, j, m, n**). #$p < 0.05$ vs. NC, *$p < 0.05$ vs.+/+. Source data are provided as a Source Data file

haplo-deficient animals, demonstrating more efficient insulin signaling (Fig. 3d).

In addition to its role in lipid storage, adipose tissue also functions as an endocrine organ secreting a large number of adipokines, which can modify systemic glucose homeostasis and insulin sensitivity. We found that circulating levels of the antidiabetic adipokine adiponectin[27] were significantly higher in

the TonEBP haplo-deficient mice on the HFD (Fig. 3e, left). Consistent with this, adiponectin mRNA expression in eWAT was higher in these animals (Fig. 3e, right). On the other hand, levels of serum adipokine leptin[28] were lower in these mice (Fig. 3f, left) as were leptin mRNA levels in eWAT (Fig. 3f, right). These data indicate that TonEBP deficiency led to an adipokine profile favoring insulin sensitivity.

Histological analysis revealed significantly lower lipid accumulation in brown adipose tissues and smaller adipocyte size in iWAT from the HFD-fed TonEBP haplo-deficient mice (Fig. 3g), indicating that TonEBP deficiency leads to a healthier adipocyte morphology as a result of lower triglyceride accumulation, possibly due to an increased rate of lipolysis. In addition, eWAT of these animals contained significantly fewer crown-like structures (Fig. 3h), lower macrophage infiltration (Fig. 3i), and lower expression of the inflammatory cytokine tumor necrosis factor α (TNFα) (Fig. 3j), indicating reduced inflammation. Obesity-induced inflammation is known to suppress genes involved in lipid metabolism in eWAT[29]. Suppression of genes involved in lipid metabolism was indeed observed in eWAT in response to HFD feeding (Supplementary Fig. 5b), but expression of these genes was restored in the TonEBP haplo-deficient animals (Supplementary Fig. 5c), consistent with the reduced inflammation.

Chronic exposure of mice to HFD causes hepatic steatosis and increased liver mass[30,31]. As expected, HFD-fed mice showed an enlarged liver (Fig. 3k) with large lipid droplets (Fig. 3l), and elevated circulating levels of alanine aminotransferase (ALT) (Fig. 3m), triglycerides, LDL, and cholesterol, but not free fatty acid (FFA) (Fig. 3n). All of these changes were dramatically tempered in the TonEBP haplo-deficient animals. In the obese $Lepr^{db/db}$ mice, TonEBP haplo-deficiency was associated with a smaller liver (Supplementary Fig. 5d), suggesting similar changes in this model of obesity. Collectively, the data in Fig. 3 demonstrate that TonEBP mediates the detrimental effects of obesity, such as insulin resistance, altered serum adipokine profile, dyslipidemia, and hepatic steatosis.

**TonEBP suppresses Adrb3 gene expression by DNA methylation.** Stimulation of Adrb3 signaling by sympathetic activation increases thermogenesis and HSL-mediated lipolysis, leading to reduced fat mass[32]. In a group of human subjects, we found that there was an inverse relationship between mRNA levels of *Adrb3* vs. *TonEBP* in subcutaneous adipocytes (Fig. 4a), suggesting that TonEBP suppressed Adrb3 expression. In mice, *Adrb3* mRNA levels in iWAT were suppressed by HFD and this suppression was reversed in the TonEBP haplo-deficiency (Fig. 4b). When these animals were exposed to cold, Adrb3 expression in iWAT was higher (Fig. 4c). Higher *Adrb3* mRNA (Fig. 4d) and protein (Supplementary Fig. 4a) expression were also observed in primary adipocytes differentiated from SVF of the TonEBP haplo-deficient animals. The same was observed in both basal and isoproterenol-stimulated, TonEBP-knocked-down 3T3-L1 cells (Supplementary Fig. 6a), which was reversed when TonEBP was overexpressed (Supplementary Fig. 6b). These data demonstrate that TonEBP suppresses Adrb3 expression in adipocytes.

We asked whether the enhanced expression of Adrb3 in response to TonEBP haplo-deficiency was responsible for the elevated expression of thermogenic genes (Fig. 2f, g) and activation of downstream signaling p-CREB (Supplementary Fig. 4a) and cAMP level (Supplementary Fig. 6c). The mRNA levels of thermogenic genes were elevated by TonEBP knockdown, while they were reduced by Adrb3 knockdown (Fig. 4e). Notably, the enhancement of mRNA expression by TonEBP knockdown was almost completely abolished by TonEBP/Adrb3 double knockdown (Fig. 4e). Also, TonEBP deficiency promotes thermogenic gene expression upon treatment with CL316,243 (Adrb3 specific agonist) (Supplementary Fig. 6d) suggesting Adrb3 dependence. These data demonstrate that TonEBP suppresses thermogenic genes via downregulation of Adrb3 in adipocytes.

We next investigated molecular basis of the TonEBP-mediated *Adrb3* gene suppression. First, we constructed a pGL3 luciferase reporter vector by inserting a 6-kb promoter sequence from the mouse *Adrb3* gene (Fig. 4f). TonEBP knockdown stimulated the *Adrb3* promoter-driven luciferase expression in 3T3-L1 adipocytes (Fig. 4g). Because the mouse *Adrb3* promoter contains one putative TonE (TonEBP binding sequence) (Fig. 4f), we investigated whether TonEBP bound to the TonE using electrophoretic mobility shift assay, and found that it did (Supplementary Fig. 6e). To confirm this interaction in situ, we performed chromatin immunoprecipitation (ChIP). Fragments of the *Adrb3* promoter containing the TonE were precipitated by TonEBP antibody, and this was reduced by TonEBP deficiency in 3T3-L1 cells (Fig. 4h) and primary adipocytes differentiated from SVF (Supplementary Fig. 6f) demonstrating that TonEBP bound to the region on the chromatin.

To investigate whether the stimulation of the *Adrb3* promoter activity in response to TonEBP knockdown was dependent on the TonE in the promoter, we constructed a TonE-deleted *Adrb3* promoter vector. The deletion mutant (ΔTonE) showed enhanced transcriptional activity, which was not affected by TonEBP knockdown (Fig. 4i), confirming functionality of the TonE on Adrb3 expression. Next, we examined chromatin accessibility in the promoter region by analyzing sensitivity to micrococcal nuclease (MNase). Chromatin accessibility was enhanced by TonEBP knockdown at both the TonEBP binding region (A in Fig. 4f) and two regions near the transcription start site (TSS, B and C) (Fig. 4j). In addition, recruitment of RNA polymerase II to the *Adrb3* promoter was also stimulated by TonEBP knockdown (Supplementary Fig. 7a) demonstrating that TonEBP binding to the TonE site led to a blockage of accessibility to chromatin around the transcription start site.

To understand molecular mechanism for chromatin remodeling, we investigated epigenetic changes in response to TonEBP deficiency. Monomethylation (H3K4me1) and trimethylation of H3K4 (H3K4me3) and acetylation of H3K27 (H3K27ac) were elevated, while trimethylation of H3K27 (H3K27me3) was reduced by TonEBP knockdown at both the TonEBP binding (A) region and region near TSS (B) of the *Adrb3* promoter consistent with activation of transcription (Fig. 4k). On the other hand, these epigenetic marks at the *GAPDH* promoter were not changed by TonEBP knockdown (Supplementary Fig. 7b). Mice with TonEBP haplo-deficiency also showed similar epigenetic changes at the *Adrb3* promoter (Supplementary Fig. 7c).

Since DNA methylation was associated with insulin resistance and thermogenic gene regulation[33,34] and could regulate histone modifications[35–37], we analyzed DNA methylation in CpG sites in the A, B, and C regions on the *Adrb3* promoter (Fig. 4f) using bisulfite sequencing (see Supplementary Fig. 8a). We found that basal levels of DNA methylation were much higher in 3T3-L1 cells compared to iWAT and primary adipocytes (Fig. 4l, Supplementary Fig. 8b and c). Consistent with this, we also found that *Adrb3* mRNA expression is 50–100 times higher in iWAT and primary adipocytes (Supplementary Fig. 8d) indicating that 3T3-L1 cells had much more of fibroblastic characteristics. TonEBP deficiency in 3T3-L1 reduced DNA methylation of the A, B, and C regions in 3T3-L1 adipocytes (Fig. 4l). TonEBP deficiency also reduced DNA methylation of the B and C regions of mouse iWAT and primary adipocytes differentiated from SVF (Supplementary Fig. 8b, c). These results demonstrate that TonEBP promotes DNA methylation of the *Adrb3* promoter in adipocytes of mice.

We found that in human subcutaneous adipocytes DNA methylation of the *Adrb3* promoter correlated negatively with *Adrb3* mRNA expression (Supplementary Fig. 8e), while correlated positively with BMI (Supplementary Fig. 8f). These data suggest that TonEBP may promote DNA methylation of the *Adrb3* promoter also in human adipocytes in view of the inverse relationship between mRNA levels of *Adrb3* vs. *TonEBP* (Fig. 4a).

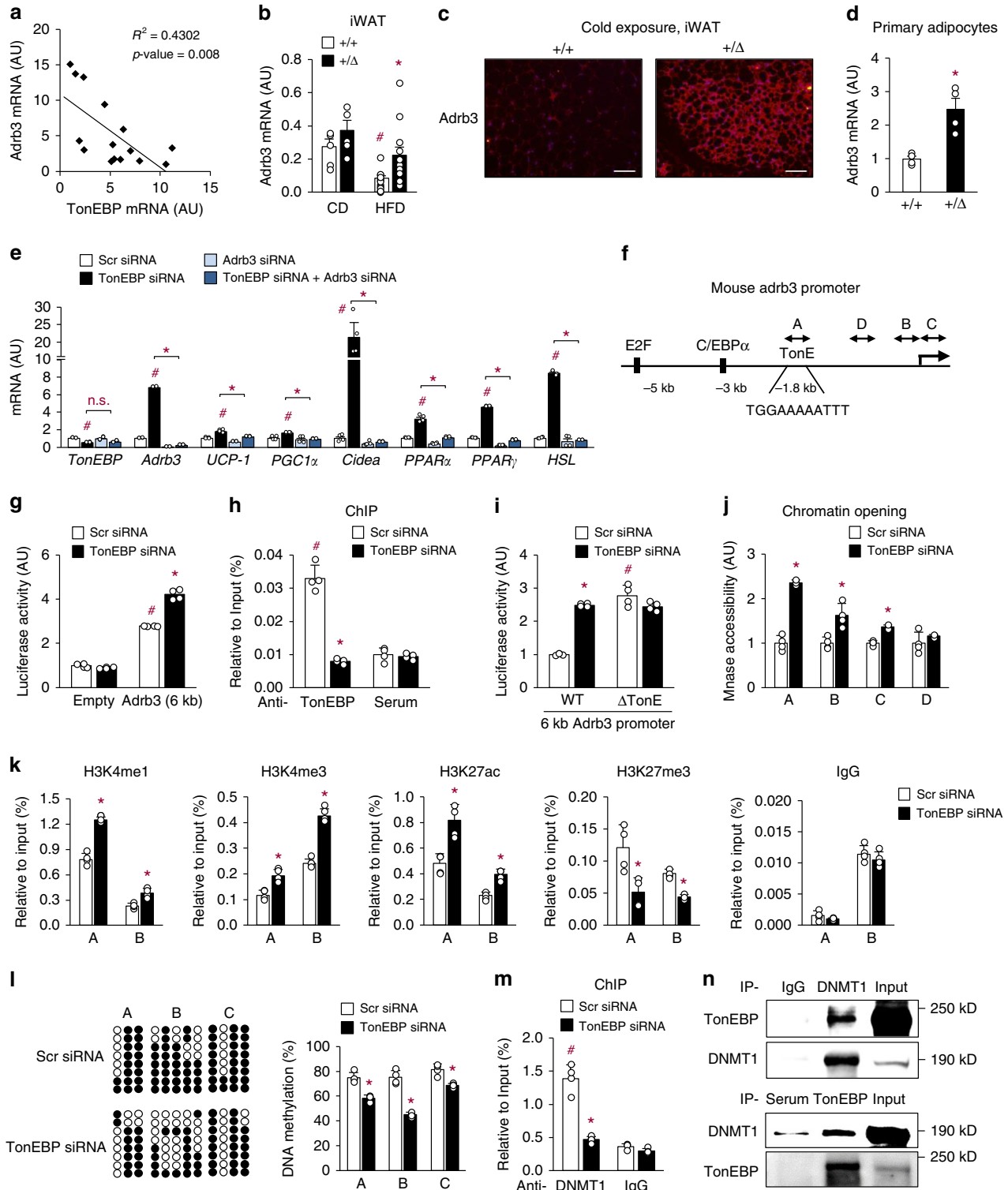

**DNMT1 inhibition enhances Adrb3 expression and beiging of WAT.** Among the known DNA methyltransferases, DNMT1 promotes insulin resistance by mediating DNA methylation in HFD-induced obese mice[38]. ChIP analysis showed that adipocyte DNMT1 was recruited to the *Adrb3* promoter in a TonEBP-dependent manner in 3T3-L1 cells and primary adipocytes (Fig. 4m and Supplementary Fig. 9a). In addition, co-immunoprecipitation experiments showed that TonEBP and DNMT1 interacted with each other (Fig. 4n). This interaction appeared to take place in the *Adrb3* promoter region because

binding of both TonEBP and DNMT1 to the promoter was TonEBP-dependent in the A region (Supplementary Fig. 9b, c). The same interaction was also observed in the B and C regions suggesting that TonEBP–DNMT1 complex bound to the A region made also contacts with the B and C regions. These findings suggest that TonEBP is responsible for the recruitment of DNMT1 to the *Adrb3* promoter leading to DNA methylation and suppression of the promoter.

To elucidate the potential role of DNA methylation in WAT beiging, we used a DNMT inhibitor, RG108. Treatment of

**Fig. 4** TonEBP suppresses *Adrb3* gene expression. **a** Correlation of *TonEBP* mRNA with *Adrb3* mRNA in human subcutaneous adipocytes ($n = 15$). **b** *Adrb3* mRNA levels in iWAT from animals fed with a CD or HFD (CD, $n = 5$; HFD, $n = 13$). **c** Adrb3 immunostaining of iWAT from animals exposed to cold (4 °C). Scale bars, 100 μm. **d** *Adrb3* mRNA abundance in primary adipocytes ($n = 4$). **e** mRNA abundance of thermogenic genes in 3T3-L1 adipocytes transfected with various siRNA's as indicated ($n = 4$). **f** Schematic representation of the mouse *Adrb3* gene promoter including the TonEBP binding site (TonE). A–C indicate regions for bisulfite sequencing ($n = 4$). **g** 3T3-L1 adipocytes were transfected with siRNA followed by an *Adrb3* promoter-luciferase reporter construct. Luciferase was measured ($n = 4$). **h** TonEBP binding to the TonE (A region) of the *Adrb3* promoter in 3T3-L1 adipocytes ($n = 4$). **i** The *Adrb3* promoter-luciferase reporter described above (WT) and a mutant reporter where TonE was removed (ΔTonE) were analyzed in 3T3-L1 adipocytes ($n = 4$). Chromatin accessibility of A, B, C, and D regions (see **f**) (**j**), and ChIP assay for H3K4me1, H3K4me3, H3K27ac, H3K27me3 and normal rabbit IgG (**k**) for A and B regions, on the *Adrb3* promoter region of 3T3-L1 adipocytes ($n = 4$). **l** DNA methylation analysis of the *Adrb3* promoter using bisulfite sequencing in 3T3-L1 adipocytes. Left: bacterial clones without (open circles) or with methylation (solid circles) from representative experiments are shown. Right: % of DNA methylation, mean + s.d. ($n = 4$). **m** ChIP assay for DNMT1 on the *Adrb3* promoter in 3T3-L1 adipocytes ($n = 4$). **n** Co-immunoprecipitation analyses of TonEBP and DNMT1 in 3T3-L1 adipocytes. *n* represents number of biologically independent samples (**a**, **b**, **d**) or independent experiments with triplicate (**e**, **g–m**). All data are presented as mean + s.e.m. (**b**, **d**) or + s.d. (**e**, **g–m**). AU arbitrary unit. The *p*-values are determined by unpaired *t*-test (**d**, **j–l**) or one-way ANOVA (**b**, **e**, **g**, **h**, **m**). #$p < 0.05$ vs. CD (**a**, **b**), scr siRNA (**e**), empty vector (**g**), anti-rabbit serum (**h**), WT (**i**), or anti-rabbit IgG (**m**). *$p < 0.05$ vs. +/+ (**a–d**), TonEBP siRNA (**e**), or scr siRNA (**g–m**). Source data are provided as a Source Data file

adipocytes with RG108 enhanced UCP-1 and Adrb3 expression in a dose-dependent manner, without affecting expression of TonEBP (Fig. 5a). When mice were treated with RG108 (12 mg/kg intraperitoneally every 2 days), they resisted HFD-induced gain in body mass (Fig. 5b, c), fat mass (Fig. 5c, d), and rise in fasting glucose (Fig. 5e). The RG108-injected mice displayed higher energy expenditure (Fig. 5f–h), resistance to cold (Fig. 5i), and enhanced beiging of WAT (Fig. 5j, k). Since there are three isoforms of DNMT—DNMT1, DNMT3a, and DNMT3b—we asked which isoform was responsible for the effects of RG108. When each isoform was individually knocked down using specific siRNA, knockdown of DNMT1 led to elevated expression of *UCP-1* and *Adrb3*, while knockdown of other isoforms did not (Fig. 5l). *UCP-1* mRNA expression was not further elevated by TonEBP/DNMT1 double knockdown (Supplementary Fig. 9d) indicating that TonEBP and DNMT1 were in the same pathway. In addition, activity of the *Adrb3* promoter (WT) was enhanced by DNMT1 knockdown (Fig. 5m, left), but that of the ΔTonE mutant promoter (Fig. 4i) was not (Fig. 5m, right). These data are consistent with the TonEBP-dependent recruitment of DNMT1 to the *Adrb3* promoter described above (Fig. 4m). Thus, TonEBP–DNMT1 complex (Fig. 4n) on the *Adrb3* promoter suppresses the promoter by DNA methylation.

**Adipocyte-specific TonEBP deficiency displays phenotypes similar to the TonEBP haplo-deficiency.** In order to directly evaluate the role of TonEBP in adipocytes, we generated adipocyte-specific TonEBP knockout mice (AKO) using the Cre-lox system (*TonEBP*^fl/fl^; adiponectin promoter-driven Cre). As controls, floxed TonEBP mice that did not express Cre recombinase were used (*TonEBP* ^fl/fl^; WT). AKO mice and their WT littermates were fed HFD for up to 12 weeks, starting at 8 weeks of age. The AKO mice were resistant to the development of obesity and showed lower fasting glucose levels than the WT mice despite no differences in food intake (Fig. 6a). In addition, the AKO mice demonstrated higher $O_2$ consumption and $CO_2$ production rates (Fig. 6b, c) without changes in RER (Supplementary Fig. 10a) and generated more heat (Fig. 6d), along with higher body temperature than the WT animals (Fig. 6e and Supplementary Fig. 10b) demonstrating higher energy expenditure. In addition, they exhibited improved glucose and insulin tolerance indicating higher insulin sensitivity (Supplementary Fig. 10c). These mice also showed higher mRNA expression of *Adrb3* and thermogenic genes and beiging phenotype in iWAT with multi-locular lipid droplet, similar to the haplo-deficient mice, when either HFD-fed (Fig. 6f) or cold-exposed (Fig. 6g, h). These data further confirm that adipocyte TonEBP suppresses thermogenesis and beiging of WAT.

## Discussion

We previously reported that TonEBP inhibited adipocyte differentiation by suppressing PPARγ2 expression[17] consistent with the low TonEBP expression in mature adipocytes. Here we discover that TonEBP expression in subcutaneous adipocytes is dramatically escalated in response to high calorie intake in two different mouse models of obesity, and obese humans. This elevation of TonEBP expression is mediated in part by a fall in miR-30 expression. TonEBP binds to its cognate DNA sequence in the *Adrb3* promoter and recruits DNMT1 DNA methylase (Fig. 6h). Ensuing DNA methylation of the promoter leads to diminished Adrb3 expression and blockade of thermogenesis and lipolysis or beiging, which in turn facilitates obesity and associated pathological states such as insulin resistance and dyslipidemia. Thus, TonEBP is a newly recognized molecular regulator, which is activated by the stress of excess calorie. Of note, relatively modest inhibition of TonEBP, i.e., TonEBP haplo-deficiency, is highly effective in preventing the pathological effects of TonEBP. As such, it is an attractive therapeutic target for obesity and associated diseases.

Inhibition of DNA methylation promotes adipogenesis[34] and improves insulin resistance[33] in obese mice much like TonEBP deficiency described here. In order to further explore the function of the TonEBP–DNMT1 pathway in adipocytes, we analyzed expression of a panel of adipocyte physiology-related genes in 3T3-L1 cells after separate knockdown of TonEBP and DNMT1 (Supplementary Fig. 11). The results indicate that more than 60% of the genes are regulated by the TonEBP–DNMT1 pathway suggesting that many aspects of adipocyte physiology are potentially affected by the pathway in addition to beiging and thermogenesis.

In mice and humans, Adrb3 stimulation increases thermogenesis, fat oxidation and lipolysis[39–42]. Adrb3 agonists also exert potent anti-diabetic effects by improving insulin sensitivity in mouse models of type 2 diabetes[39,43]. Moreover, in humans Adrb3 stimulation induces thermogenesis with beneficial metabolic effects[44]. In adult humans, on the other hand, white adipose tissue is characterized by a low expression of the Adrb3[45]. This limitation may explain the lack of efficacy by Adrb3 agonists in clinical trials[46]. Therefore, boosting the Adrb3 expression by means of TonEBP inhibition offers distinctive opportunity in therapeutic approach for obesity and metabolic diseases.

TonEBP was originally identified as a DNA binding transcriptional enhancer[5]. It is clear now that this function applies mostly to genes stimulated by hypertonicity such as aldose reductase or the sodium/myo-inositol cotransporter. In macrophages, TonEBP stimulates a host of pro-inflammatory genes as a transcription cofactor for histone acetylation of the promoters without DNA binding[10]. Likewise, in adipocytes and macrophages, TonEBP

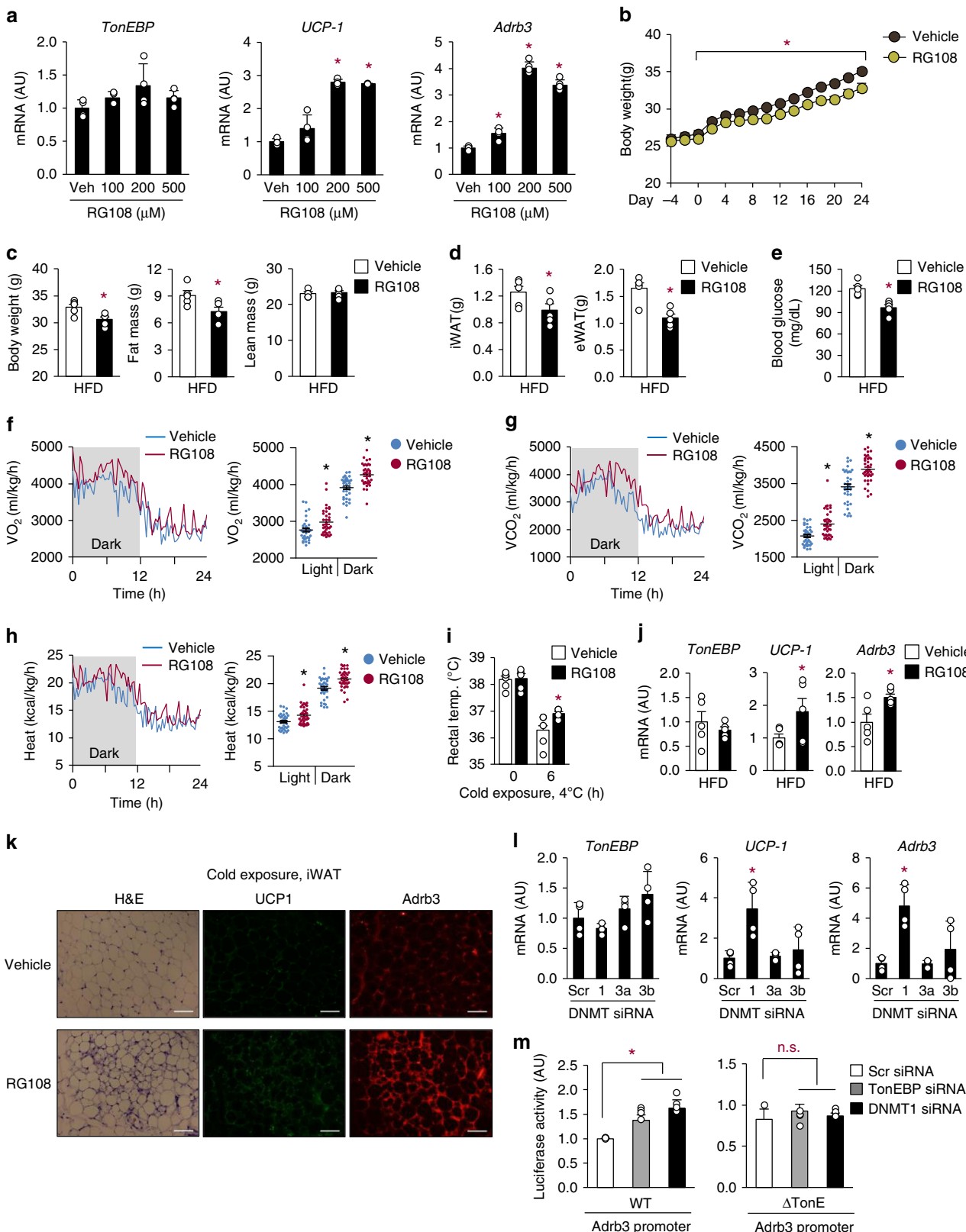

suppresses transcription of PPARγ2 by mediating histone methylation of its promoter region without DNA binding[17]. Here we add transcriptional suppressor function of TonEBP that involves DNA binding and DNA methylation. When TonEBP is knocked down in various cultured cells, typically over 500 genes show changes in mRNA expression in such a way that half of them increase, while

the other half decrease[13]. Such pleiotropic functions of TonEBP—both transcriptional stimulation and transcriptional suppression—are consistent with its bidirection regulation of many genes.

In macrophages TonEBP is a critical regulator of M1 polarization, and TonEBP haplo-deficient animals display a dramatically reduced inflammation in settings of various inflammatory

**Fig. 5** DNMT suppresses WAT beiging and Adrb3 expression. **a** *TonEBP*, *UCP-1* and *Adrb3* mRNA levels in 3T3-L1 adipocytes treated with 0–500 μM RG108 as indicated for 24 h ($n = 4$). **b–e** Animals were treated with vehicle or RG108 every 2 days from day −4 to 24. All animals were switched from CD to HFD on day 0 ($n = 5$). Body weight (**b**), body composition by echo MRI (**c**), fat pad mass (iWAT and eWAT) (**d**), and fasting blood glucose levels (**e**) were measured. **f–k** Animals were treated as above, except that the experiments were terminated on day 8 ($n = 4$). VO$_2$ (**f**), VCO$_2$ (**g**), heat production (**h**), and rectal temperature (**i**) were measured. *TonEBP*, *UCP-1*, and *Adrb3* mRNA (**j**) were measured, and H&E and immunostaining for UCP-1 and Adrb3 (**k**) were performed from iWAT. Scale bars, 100 μm. **l** *TonEBP*, *UCP-1*, and *Adrb3* mRNA levels in 3T3-L1 adipocytes transfected with scr siRNA, DNMT1 (1), DNMT3a (3a) or DNMT3b (3b) siRNA ($n = 4$). **m** 3T3-L1 adipocytes transfected with siRNA were transfected a second time with WT (left) or ΔTonE (right) *Adrb3* promoter-luciferase reporter. Luciferase activity was measured ($n = 4$). $n$ represents number of independent experiments with triplicate (**a**, **l**, **m**) or biologically independent samples (**b–j**). All data are presented as mean + s.d. (**a**, **l**, **m**) or s.e.m. (**e**, **g–m**). AU arbitrary unit. The $p$-values are determined by unpaired $t$-test (**c–e**, **j**) or one-way ANOVA (**a**, **b**, **f–i**, **l**, **m**). *$p < 0.05$ vs. 0 (**a**), vehicle (**b–j**), or scr siRNA (**l**, **m**). Source data are provided as a Source Data file

---

diseases including atherosclerosis[15], rheumatoid arthritis[13,14], and diabetic nephropathy[12] in association with reduced macrophage activity. Since these diseases are also associated with obesity, insulin resistance, and dyslipidemia[47], TonEBP inhibition would provide a multipronged therapeutic approach to a host of complicated metabolic and inflammatory diseases.

## Methods
**Mice**. All the methods involving live mice were carried out in accordance with the approved guidelines. All experimental protocols were approved by Institutional Animal Care and Use Committee of the Ulsan National Institute of Science and Technology (UNISTACUC-12-15-A).

All studies used male C57BL/6 J background mice. TonEBP haplo-deficient mice (*TonEBP* + /Δ) were described previously[5]. Mice carrying the loxP-targeted *TonEBP* gene (*TonEBP$^{fl/fl}$*) were provided by Dr Neuhofer[48]. Leptin receptor mutant *db/db* (*Lepr$^{db/db}$*) mice and *Adiponectin-Cre* transgenic mice (*AQ-cre*) were obtained from The Jackson Laboratory (USA). *TonEBP$^{fl/fl}$* mice were bred with *AQ-Cre* knock-in mice to generate *TonEBP$^{fl/fl}$* *AQ-Cre* mice.

For obese mice model, mice at 8 weeks age were fed either a CD (10% fat as kcal, Research Diets, NJ, USA) or HFD (60% fat as kcal, Research Diets) during 16 weeks. Mice were intraperitoneally injected with RG108 (Cayman, MI, USA) at 12 mg/kg every 2 days.

**Cells and reagents**. Pre-adipocyte cell line 3T3-L1 (CR-173) and HEK293 (CRL-1573) cells from American Type Culture Collection were cultured in Dulbecco's Modified Eagle's Medium (DMEM) containing 10% fetal bovine serum (Thermo Fisher Scientific Inc., MA, USA) and penicillin/streptomycin (100 U/ml and 100 μg/ml; GE Healthcare Life Sciences, UT, USA). For induction of beige adipocyte differentiation, beige adipogenesis inducing medium (BAIM), including 1 μM dexamethasone, 0.5 mM isobutylmethylxanthine, 1 μM insulin, 125 μM indomethacin, and 1 nM triiodothyronine was used[49]. Cells were maintained at 37 °C in incubator with 5% CO$_2$. Cells were treated with isoproterenol (Sigma Aldrich, USA) for 4 h after adipocyte differentiation.

**Human adipocyte samples**. Adipocyte RNA of subcutaneous abdominal adipose tissues was obtained from 15 Saudi Arabian subjects. The local ethics committee of the College of Medicine, King Saud University, approved experimental design and adipose tissue samples collection (approval code 07-602). The procedures were carried out in accordance with the approved protocol, with written informed consent obtained from all subjects. Further information and requests for resources and raw data should be directed, and will be fulfilled by the contact: A. A. Alfadda, aalfadda@ksu.edu.sa.

**Transfection**. When the cells grow at the 70% of confluence, cells were transfected for 48 h with TonEBP siRNA or control scrambled siRNA (10 nM) and miR-negative control, miR-30b or miR-30c (100 nM) using lipofectamine RNAiMAX (Invitrogen, CA, USA) following the manufacturer's instructions.

**Isolation and differentiation of preadipocytes**. Preadipocytes were isolated from iWAT obtained from 4-week-old animals. iWAT was minced in DMEM/F12 medium and digested with 10 ml HEPES buffer containing 1 mg/ml type II collagenase. They were incubated at 37 °C with gentle shaking for 1 h. After incubation, 10 ml of DMEM/F12 was added and they were centrifuged at $500 \times g$ for 15 min. Stromal vascular cells (SVCs) were incubated with 5 ml of RBC lysis buffer (Sigma Aldrich) for 5 min at room temperature and then they were filtered with a 40 μm filter after addition of 10 ml of DMEM/F12. After centrifugation at $500 \times g$ for 5 min, SVCs were differentiated to beige adipocytes using BAIM.

**Immunoblot assay**. Cells were washed two times with cold PBS and lysed in RIPA buffer (0.01 M Tris, pH 7.4, 0.15 M NaCl, 0.001 M EDTA, 0.001 M EGTA, 1%

Triton-X 100, 0.002 M PMSF, and protease inhibitor (Roche, Rotkreuz, Switzerland)). After centrifugation of lysate, supernatant was used for immunoblot assay. Protein concentration was measured by BCA protein assay system (Pierce Biotechnology, IL, USA). Equal amounts of protein from each sample were separated by SDS-PAGE. Anti-Adrb3 (Abcam, Cambridge, UK, #ab59685), anti-UCP-1 (EMD Milipore, CA, USA, #AB1426), anti-pAKT (Cell signaling, #9271 S), anti-AKT (Cell signaling, #9272), anti-Hsc70 (Rockland, PA, USA, #200-301-A28), and anti-TonEBP antibody[5] were used for immunoblotting at 1:100 dilution. HRP-conjugated mouse, rabbit, or goat secondary antibodies were used for detection. The antigen–antibody binding was detected by chemiluminescent detection reagent (GE Healthcare, NJ, USA). Uncropped western blots are shown in Supplementary Fig. 12.

**RNA isolation and real-time PCR**. Total RNA from human or mouse adipocytes, mouse adipose tissues, or 3T3-L1 cells was isolated using the TRIzol reagent (Invitrogen, CA, USA) according to the manufacturer's instructions. cDNA was synthesized by M-MLV reverse transcriptase (Promega, WI, USA). After reverse transcription, real-time PCR was performed using SYBR Green I Master and LightCycler 480 II (Roche). Measured cycle threshold values were normalized for the cyclophilin A or 36B4 reference gene, and they were expressed as fold change over control samples. Primers used are described in Supplementary Table 1.

**Immunohistocytochemistry**. Cells were grown on glass coverslips and fixed with 4% paraformaldehyde in PBS (pH 7.4) for 20 min at 4 °C. Cells were permeabilized with 0.3% Triton-X 100 in PBS for 30 min and blocked with PBS containing 3% goat serum and 1% bovine serum albumin for 1 h at room temperature. After incubation with rabbit anti-UCP-1 (EMD Millipore, #AB1426), Adrb3 (Abcam, #ab59685), and rat anti-F4/80 (Abcam, #ab6640) at 1:100 dilution overnight at 4 °C, the cells were washed with PBS and treated with goat anti-rabbit or anti-rat Alexa Fluor 488-conjugated and Alexa Fluor 594-conjugated secondary antibodies at 1:1000 dilution for 1 h. Cells were washed with PBS and incubated in 0.1 μg/ml Hochest (DAPI) for 30 min. After washing with PBS, coverslips were mounted onto microscope slides. Images were recorded using an Olympus FV1000 confocal fluorescence microscope.

**Metabolic analysis**. Fasting blood glucose, body weight, and food intake were measured weekly, and body composition was measured using a quantitative nuclear magnetic resonance system (EchoMRI100V; Echo Medical Systems, TX, USA). Oxygen consumption, carbon dioxide production, heat production, and locomotor activity were monitored using Comprehensive Lab Animal Monitoring System (Columbus Instruments, OH, USA). Mice were given an oral injection of D-glucose (2 g/kg body weight) after overnight starvation for the glucose tolerance test and were intraperitoneally injected with insulin (0.75 U/kg body weight) for insulin tolerance test. Serum glucose levels were determined in tail blood samples using a glucometer. Body temperatures were measured using a digital thermometer (TD-300; Shibaura Electronics, Tokyo, Japan).

**Electromobility shift assay**. Cells were harvested and centrifugation at $500 \times g$. The cell pellet was washed by suspension with PBS. The cell nucleus and cytoplasm were separated by using the Nuclear and Cytoplasmic extraction kit (Pierce), according to the manufacturer's instruction. Nuclear fraction was confirmed by Lamin B.

EMSA assay was performed using Lightshift Chemiluminescent EMSA kit (Pierce). Five micrograms of nuclear extracts were incubated with poly(dI:dC), binding buffer, and 5′ biotinylated DNA (5′-CAATTTGGAAAAATTTTGACT-3′ for TonEBP binding site on *Adrb3* promoter) at room temperature for 20 min. Samples were separated by electrophoresis for 4 h in 4% (40% 29:1 acrylamide/bis solution) gel for TonEBP. The detection was performed according to the manufacturer's instructions.

**Immunoprecipitation assay**. Total cell lysates (10–500 μg) were prepared using RIPA buffer in a tube on ice. Antibody (1–5 μg) was added to cell lysate and they

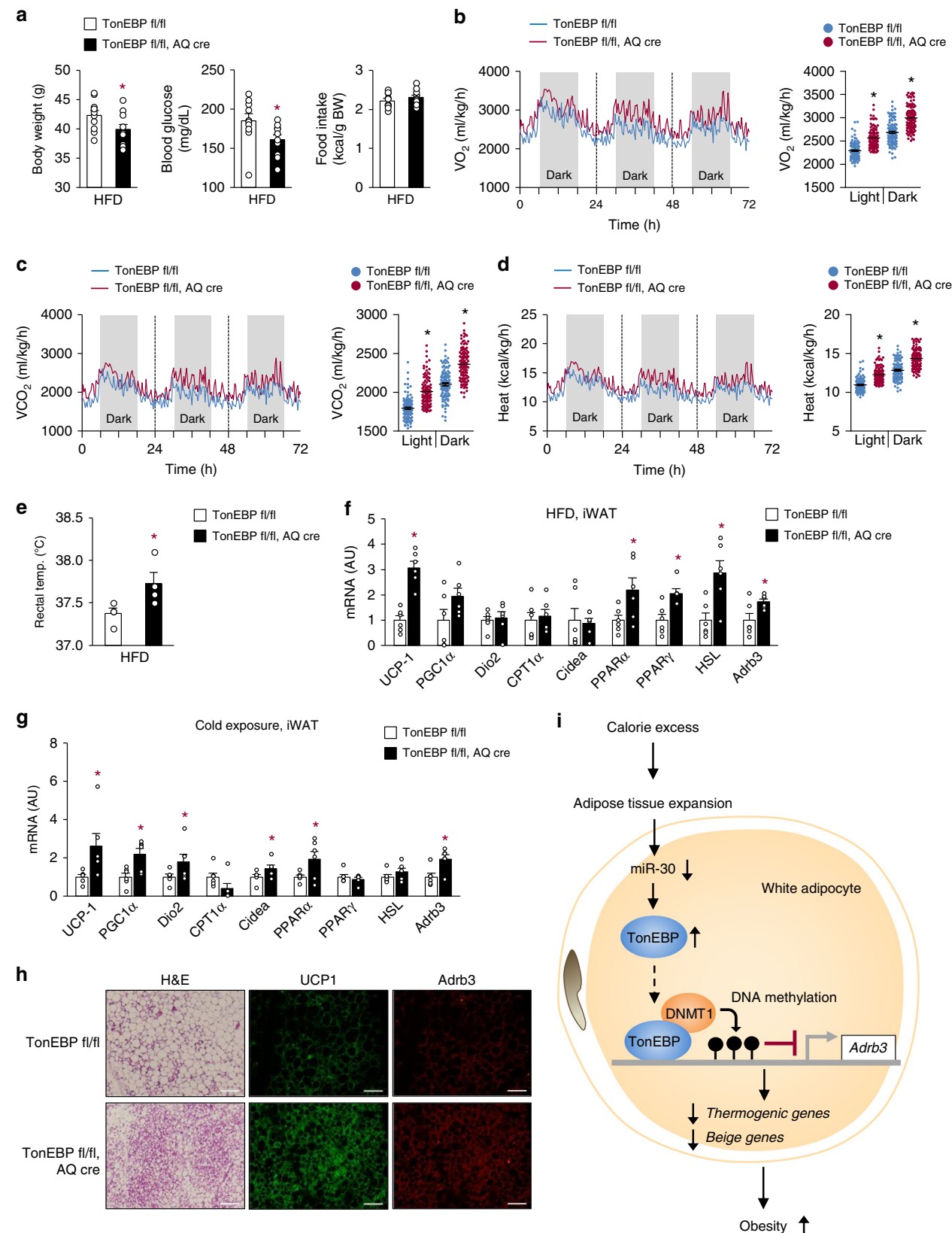

were incubated for overnight at 4 °C under rotary agitation. Forty microliters of protein A/G agarose beads (GE Healthcare) were added and incubated for 2 h at 4 °C under rotary agitation. The bead–antibody–antigen complex was spun down by centrifugation at 4 °C for 1 min and removed the supernatant. The complex was washed three times for 10 min by RIPA buffer at 4 °C. After washing, 40 μl of sample buffer was added and boiled at 95 °C for 5 min. The samples were analyzed by immunoblotting. Anti-DNMT1 (Abcam, #ab188453) and anti-TonEBP

antibody[5] were used for immunoblotting at 1:1000 dilution. Uncropped western blots of immunoprecipitated proteins are shown in Supplementary Fig. 13.

**Chromatin immunoprecipitation assay.** Cells were grown in 10-cm-diameter culture dishes. Fixation was performed with 1% formaldehyde at room temperature for 10 min. The fixation was stopped with 0.125 M glycine for 5 min at room

**Fig. 6** Adipocyte-specific TonEBP deficiency promotes energy expenditure and beiging of WAT. **a** Body weight, fasting blood glucose, and food intake by adipocyte-specific TonEBP knockout mice (*TonEBP*$^{fl/fl}$, *AQ-cre*) and their cre-negative littermates (*TonEBP* $^{fl/fl}$) fed with HFD for 12 weeks ($n = 10$). VO$_2$ (**b**), VCO$_2$ (**c**), heat production (**d**), and rectal temperature (**e**) analyzed in animals fed with HFD ($n = 4$). mRNA expression of thermogenic genes in iWAT from animals fed with HFD (**f**) or exposed to cold conditions (**g**) ($n = 4$). **h** Representative images of iWAT sections stained with H&E, UCP-1, and Adrb3 antibody from CD-fed animals exposed to cold (4 °C). Scale bars, 100 μm. **i** Proposed model for the inhibition of beiging in white adipocytes by TonEBP. TonEBP is induced by excess calorie intake. It binds to the *Adrb3* promoter where it recruits DNMT1 leading to DNA methylation and suppression of the promoter. *n* represents number of biologically independent animals (**a**–**e**) or samples (**f**, **g**). All data are presented as mean + s.e.m. AU arbitrary unit. The *p*-values are determined by unpaired *t*-test (**a**–**e**) or one-way ANOVA (**b**–**d**, **f**, **g**). *$p < 0.05$ vs. *TonEBP*$^{fl/fl}$. Source data are provided as a Source Data file

temperature. After three-times wash with cold PBS, cells were collected and lysed in 1 ml of SDS lysis buffer (1% SDS, 10 mM EDTA and 50 mM Tris-HCl pH 8.1) for 10 min on ice. Cell lysates were sonicated (Bioruptor KRB-01, BMS, Tokyo, Japan) for six cycles of 20 s on plus 30 s off with constant frequency and maximum intensity to obtain DNA fragments between 400 and 1000 bp. Each sample was diluted 1:10 with dilution buffer (0.01% SDS, 1.1% Triton-X 100, 1.2 mM EDTA, 16.7 mM Tris-HCl, pH 8.1, and 167 mM NaCl) for immunoprecipitation. Samples were precleared with protein A Sepharose beads (Millipore, MA, USA) that were previously pre-adsorbed with salmon sperm DNA for 1 h at 4 °C. Specific antibodies were added after removing the preclearing beads: anti-PolII IgG (Abcam, #ab817), anti-DNMT1 IgG (Abcam, #ab188453), anti-H3K4me1 IgG (Abcam, #ab8895), anti-H3K4me3 (Abcam, #ab8580), anti-H3K27me3 (Abcam, #ab6002), anti-H3K27ac (Abcam, #ab4729), anti-H3ac (EMD Millipore, #06–299), normal rabbit IgG (Abcam, #ab171870), anti-TonEBP serum[5], and normal rabbit serum (Merck Millipore, Darmstadt, Germany). After adding antibodies (5 μl), the lysates were incubated overnight at 4 °C. Protein A Sepharose beads were then added, incubated for 2 h at 4 °C, and then washed with low salt washing buffer (0.1% SDS, 1% Triton-X 100, 20 mM Tris-HCl pH 8.1, 2 mM EDTA, and 10 mM NaCl), high salt washing buffer (0.1% SDS, 1% Triton-X 100, 20 mM Tris-HCl, pH 8.1, 2 mM EDTA, and 500 mM NaCl), LiCl washing buffer (0.25 M LiCl, 1% NP-40, 1% deoxycholic acid, 1 mM EDTA, and 10 mM Tris-HCl, pH 8.1) and twice with final washing buffer (10 mM Tris-HCl, pH 8.0 and 1 mM EDTA). To elute the DNA, beads were incubated with elution buffer (1% SDS and 100 mM NaHCO$_3$) for 20 min at 65 °C. To reverse the cross-linking, samples were incubated overnight at 65 °C in 200 mM NaCl, 30 min at 37 °C with 50 μg/ml RNase (Pierce) and 2 h at 45 °C with 100 μg/ml proteinase K. DNA was purified using the PCR purification kit (Qiagen, CA, USA). DNA was then subjected to real-time PCR using primers: 5′-GACAACTCATGGAGCAGTCTT-3′ and 5′-CTTACTTACTGTGCCATCT CCC-3′ for the A region, 5′-GCCTTCCTTGTTGGGTAAAGGATA-3′ and 5′-CAGCCTGGGCCATCTTCCCTAATTG-3′ for the B region, 5′-GAGGGGGAA CCTTCCCACCCCA-3′ and 5′-AGTCCCACTACCAAGTCAGCTGCG-3′ for the C region, and 5′-GGAGGAAGATGGAAACCAGAAG-3′ and 5′-CGCCACTATT CCCAACTCTAAG-3′ for the D region of mouse *Adrb3* promoter; 5′-GGGTTC CTATAAATACGGACTGC-3′ and 5′-CTGGCACTGCACAAGAAGA-3′ for the region of mouse GAPDH promoter. Immunoprecipitated DNA from each sample was normalized to its respective chromatin input.

**Luciferase assay.** Cells were transfected with either a TonE-driven Photinus luciferase or an *adrb3* promoter 6kb-driven luciferase reporter vector (pGL3-promoter vector, Promega). The Renilla luciferase reporter plasmid (pRL-TK, Promega) was used as a control for transfection efficiency. To verify the binding of miR-30b and miR-30c to the predicted sites in the 3-UTR region of the TonEBP gene, predicting algorithms in Targetscan (www.targetscan.org) and psiCHECK-2 vector were used. A three-hundred base-pair region of the predicted miR-30b and miR-30c binding sites in 3′-UTR of the TonEBP gene was cloned into a psiCHECK-2 (Promega) downstream of the Renilla luciferase-coding region. Luciferase activity was measured using the Dual-Luciferase Assay System (Promega) according to the manufacturer's instructions. Luciferase activity was normalized by activity of renilla luciferase.

**ELISA and lipid analysis.** Leptin and adiponectin in serum from mice were analyzed by ELISA using a commercial kit (R&D Systems, MN, USA). Insulin was analyzed by ELISA using a commercial kit (Alpco, NH, USA).

Triacylglycerol (TG), FFA, and cholesterol level in the serum were measured using TG, FFA, or cholesterol quantification kit (Abcam) according to the manufacturer's instructions.

**MNase accessibility assay by real-time PCR.** MNase accessibility assay by real-time PCR was performed using MNase[17]. Washed cells were lysed in cold NP-40 lysis buffer (10 mM Tris-HCl (pH 7.4), 10 mM NaCl, 3 mM MgCl$_2$, 0.5% NP-40, 0.15 mM spermine (Sigma Aldrich), and 0.5 mM spermidine (Sigma Aldrich), followed by incubation on ice for 5 min. Nuclei were pelleted by centrifugation at 2500 × *g* for 3 min at 4 °C and resuspended with MNase digestion buffer without CaCl$_2$ (10 mM Tris-HCl (pH 7.4), 15 mM NaCl, 60 mM KCl, 0.15 mM spermine, and 0.5 mM spermidine). After centrifugation, the nuclei were resuspended with MNase digestion buffer supplemented with CaCl$_2$. Then, half of the each sample was treated MNase with 5 unit/sample

and the other half of that was treated digestion buffer, and samples were incubated at 37 °C for 1 min. The digestion reaction was stopped by addition of stop solution (100 mM EDTA/10 mM EGTA (pH 8.1) in 10 mM Tris-HCl (pH 7.4)). RNaseA (10 μg/sample) and proteinase K (100 μg/sample) were added and samples were incubated at 37 °C overnight. DNA, purified by phenol/chloroform/isoamyl alcohol extraction, was analyzed by real-time PCR using primer pairs covering near −1.8-kb region of the *Adrb3* promoter. Chromatin accessibility was calculated from (amount of PCR product in undigested sample)/(amount of PCR product in digested sample).

**DNA methylation analysis using bisulfite sequencing.** DNA was purified by DNA purification kit (Qiagen). Bisulfite conversion was performed using EpiTect Bisulfite Kit (Qiagen). After conversion, *Adrb3* promoter region of mouse or human was amplified by real-time PCR using primers designed by Methprimer software (www.urogene.org/methprimer/index1.html). Amplified DNA was cloned into bacteria by TOPO TA cloning kit (Invitrogen). Plasmid from each clone was sequenced by Macrogen commercial services (Daejeon, Korea).

**Statistical analysis.** Data are presented as means + s.d. or + s.e.m. Statistical significance ($p < 0.05$) was estimated by an unpaired *t*-test for comparisons between two conditions. A one-way ANOVA was used for comparisons between more than two conditions. Tukey's post hoc test was used for multiple comparisons. All statistics were performed with GraphPad Prism 5.0 software (GraphPad, CA, USA).

**Reporting summary.** Further information on research design is available in the Nature Research Reporting Summary linked to this article.

## Data availability
Data supporting the findings of this study are available from the corresponding authors upon reasonable request. The source data underlying Figs. 1–6 are provided as a Source Data file.

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

## Acknowledgements

This work was supported by the National Research Foundation grants (NRF-2018R1A5A1024340, 2017R1E1A1A074673, and 2016R1D1A1B03932335) and Health Technology R&D Project grant (HI16C1837) of Korea. This work was also supported by UNIST fund (1.170010.01).

## Author contributions

H.H.L., S.Y.C., and H.M.K. designed the experiments and wrote the paper. H.H.L., S.Y.C., S.M.A., B.J.Y., J.H.L., E.J.Y., G.W.J., H.J.K., and S.W.L performed the experiments. H.H.L., S.Y.C., H.M.K., W.L., A.A., K.J., and P.J. analyzed the data.

## Additional information

**Competing interests:** The authors declare no competing interests.

