## [Peer Review File · Nature Communications]

Reviewers' comments:

Reviewer #1 (Remarks to the Author):

In the present study entitled 'TonEBP//NNFAT5 promotes obesity and insulin resistance by epigenetic suppression of white adipose tissue beiging', Lee et al. identify TonEBP as a repressor for some thermogenic genes.

As I am no expert in the physiology of fat tissue and obesity, I will comment only on the epigenetic aspect of this story. In general, the idea that TonEBP might regulate gene expression of thermogenic genes by recruitment of DNMT1 and subsequent DNA methylation seems straightforward. However, the model for this epigenetic regulation is still underdeveloped and require additional experimentation.

Specific points:

The study tries to link TonEBP-mediated recruitment of DNMT1 as a silencing mechanism of Adrb3. The model presented in Figure 6H proposes that this is a general mechanism to silence thermogenic/beiging genes. Given that the authors propose that TonEBP might be a potential therapeutic target it seems reasonable to probe this on a genome-wide scale. One way to address this would be to perform RNA-seq and compare the overlap of misregulated genes in TonEBP and DNMT1 knock-downs (knock-outs?).

Figure 4

The ChIPs should be compared to an independent locus to probe for differences in the IP. What is the standard deviation of the ChIP? There should at least be 3 independent, biological replicates! The effect on DNA methylation in the absence of TonEBP is very modest, particularly at positions A (TonEBP binding site) and C. This raises the question of how TonEBP recruits DNMT1 1.8kb further down at the core promoter (position B) to change/maintain DNA methylation? Generally, the TonEBP binding site is fairly far away (1.8kb) to be part of a promoter region (although not impossible). The question arises if this region is more of a proximal poised enhancer than promoter. In this case, TonEBP might loop to the core promoter element. This might explain the more prominent effect on the core promoter. Experiments to be done should address if TonEBP is also recruited closer to TSS. The epigenetic landscape of this genomic region should be characterised in more detail, including ChIPs for H3K4me3, H3K27me3, H3K27ac and H3K4me1. Additionally, given the modest effects of DNA methylation in the absence of TonEBP, is forced recruitment of DNMT1 (e.g. by inactive Cas9) in the absence of TonEBP sufficient to keep Adrb3 expression low?

In contrast to the high methylation rate in NIH3T3-L1s (Figure 4), the methylation levels in vivo are relatively low (around 30% - Figure S12B) – is there an additional pathway regulating Adrb3? The authors should at least comment on this point!

General points:

miRNA 30 is introduced very rapidly – Has there been a link with TonEBP before? It should maybe be introduced shortly in the introduction.

Please always indicate in the legends what cell types have been used for the experiments, as different systems are used for different experiments and in the present form it is sometimes difficult to understand whether primary tissues/cells or cell lines were used. I suppose that it's a heterozygous knock-out mouse that was used in Fig S12B?

Reviewer #2 (Remarks to the Author):

In this work Tonicity-responsive Enhancer Binding Protein (TonEBP) is presented as an epigenetic regulator leading to obesity and insulin resistance. Its expression is upregulated in murine and human models of obesity, resulting in suppression of thermogenic capacity and of browning in

subcutaneous white adipose tissue. TonEBP role in the context of obesity was investigated using TonEBP haplo-deficient mice fed chow or high fat diet and TonEBP knock-down cultured adipocytes. Authors demonstrate that decreased TonEBP expression leads to reduced body weight, increased energy expenditure and thermogenesis, ameliorated glucose metabolism and insulin resistance. TonEBP increases DNA methylation on the promoter of beta-3 adrenoreceptor gene (*Adrb3*), by recruiting DNMT1 DNA methylase, thereby negatively regulating thermogenic and browning gene expression. Furthermore, a link between TonEBP and miR30 is reported.

Based on the results showed, TonEBP appears as an interesting and novel target linking adipose tissue metabolism, inflammation and systemic alterations in obesity and epigenetic regulation.

I have the following concerns regarding this manuscript:

Main concerns:

1. 3T3-L1 knock-down adipocytes do not completely recapitulate the in vivo effects. For example, UCP1 expression is increased in iWAT and SVF from haplo-deficient mice while this is not observed in 3T3-L1 knocked-down cells. Moreover, the expression of some of the measured genes is not strongly increased, raising some doubts about biological relevance of these differences. Figure 4E shows 20-fold induction of *Cidea* while this is not observed in Figure S6A, even though experimental conditions are similar (Con-Scr siRNA vs Con-TonEBP siRNA in Fig S6A and Scr siRNA vs TonEBP siRNA in Fig. 4E). For the above-mentioned reasons, I doubt this in vitro model could be the more appropriate to study TonEBP regulation. Authors should consider repeating some experiments in another adipocyte cell line. In particular, they should verify experiments in Figure S6A and 4E. Similarly, it would be greatly appreciable if authors could confirm TonEBP ChIP results shown in Fig 4H either in an alternative adipocyte cell line or in vivo in iWAT.

2. TonEBP is overexpressed to evaluate its impact on thermogenic gene expression (Figure S6B). However, TonEBP mRNA levels are not significantly increased, thereby no conclusive explanation on observed suppression of thermogenic genes is possible. Since mRNA levels did not increase much upon overexpression of TonEBP, authors should check whether Adenovectors increased TonEBP protein levels.

3. Authors analysed all data using Student's t-test. However, if groups are more than two, Student's t-test is not the most appropriate and suitable statistical test. Standard deviation is missing in the Figures 4H, K, L, M, S12B. Moreover, authors did not report any statistical significance in Figures 4H, K, L, M.

4. Authors did not rule out that increased thermogenesis could be due to possible effects of the haplo-deficiency of TonEBP in BAT. They should measure gene expression in BAT to provide information about BAT contribution on the observed phenotype.

5. Basal temperature and cold exposure were carried out in CD-fed mice (Figure 2D and E). Considering the role of TonEBP in response to HFD, it would be a stronger evidence to perform these experiments also in HFD setting.

6. The WB shown in Figure 1B does not include a reference control protein to verify equal loading. Could you please add it?

Minor concerns:

1. Quantification is missing in all western blots. WB quantification by densitometry should be reported.

2. Figure 1A: Due to high expression of TonEBP in iWAT in mice on HFD, the scale does not allow to appreciate the expression in other tissues. Therefore, I recommend to show data from iWAT and from other tissues separately in different panels to better emphasize differences also in other tissues.

3. Figure 1E: The WB presents two samples per group while n=4 is reported in the caption. Could you clarify this point?

4. Figure 1I: In the sentence at page 6 "Examination of dissected fat tissues confirmed that epididymal, inguinal, and dorsal fat pads were smaller in these animals" the word "fat" is misspelled ("fad" instead of "fat") and should be corrected. Furthermore, this figure shows the reduction of adipose tissue mass while Figure S4 reports the absence of alterations in food intake. In the first line of the paragraph "Adipocyte TonEBP suppresses thermogenesis and beiging of WAT" (page 6, line 16), the reference to Figure S4 where you mention food intake in the text should be added to the reference to Figure 1I.

5. Figure 2A,B, 6B,C: Authors should calculate and report the Respiratory Exchange Ratio.

6. Figure 2D and E: The basal body temperature is different between the two figures. Furthermore, at time 0 of cold exposure the difference between the two groups is not statistically significant whereas it is in Figure 2D. How would you explain this discrepancy?

7. Figure 2H and I: The diet used in this experiment should be reported.

8. Figure 2J and K: At pg. 7, authors stated in the text "In adipocytes differentiated from SVF of the TonEBP haplo-deficient animals or 3T3-L1 cells whose TonEBP was knocked down showed elevated basal and isoproterenol-stimulated expression of thermogenic genes...". This sentence is ambiguous as it sounds like both adipocytes from SVF and 3T3-L1 adipocytes were treated with isoproterenol, while this does not seem the case (i.e., adipocytes from SVF were not treated with isoproterenol, while 3T3-L1 adipocytes were treated with isoproterenol).

9. Figure 3D, E, F: Authors should provide a rationale whereby the analyses in these panels were performed in eWAT and not iWAT. Is this because eWAT is more relevant in potentially affecting insulin signalling and adipokine secretion owing to inflammation induced by HFD?

10. Figure 3G: The bar scale should be reported indicating the unit.

11. Figure 3N: Could you uniform the unit $\mu\text{g}/\mu\text{L}$ of total cholesterol to mg/dL?

12. Figure 4H, K, M: Authors should report enrichment in a DNA negative control region to show that enrichment is specific for the considered chromatin region.

13. Figure 4L: This panel is not clear at all. Please, provide a clear explanation of the figure either in the text or in the figure legend.

14. Materials and methods: Mice. Please specify the mice sub-strain (C57BL/6J or C57BL/6N?). Cells and reagents: Authors stated that they used RAW264.7 cells, but no data with this cell line are shown. Chromatin Immunoprecipitation assay: Authors stated that cells were grown with LPS. This is probably a typing error.

15. Figure S1B: TonEBP mRNA expression was elevated in iWAT from 16-week-old db/db obese mice compared to their lean db/+ littermates (page 5). In the caption is reported that mice were 10-week-old. The age of these mice should be reported consistently in the text and in figure caption.

16. Figure S13: These correlations are not totally conclusive. Authors should tone down the conclusion in the text stating "These results demonstrate that TonEBP promotes DNA methylation of the *Adrb3* promoter in adipocytes of mice and humans". The graph in Figure S13A only shows a negative correlation between DNA methylation of the *ADRB3* promoter and the levels of *ADRB3* mRNA in human subWAT. Nonetheless, this does not represent the proof that TonEBP promotes DNA methylation of *ADRB3* promoter in humans: it's only a correlation that could suggest a possible role of TonEBP in human WAT physiology speculating on the observations in TonEBP+/- mice. Also, note that human gene symbols should be upper case (i.e., *ADRB3*, not *Adrb3*).

Reviewer #3 (Remarks to the Author):

In this article, Lee et al. seek to investigate the role of transcriptional regulator TonEBP in beiging of adipose tissue and the functional consequences on systemic metabolism. The authors used both whole body and adipose tissue specific KO as well as pharmacological compounds to tackle various aspects of adipose metabolism and TonEBP biology. While TonEBP-regulation of the *Adrb3* promoter through *Dnmt1* was shown convincing, serious weaknesses exist in causally linking that single event to the systemic effects observed in the animals. Furthermore, as TonEBP has been previously shown to transcriptionally control adipocyte differentiation and subsequently insulin sensitivity, I'm concerned about the significance of the conceptual advance offered by this work in its current form. In order to be appropriate for publication in Nature Communications, I believe the following points should be addressed.

Major Points:

1. What was the rationale for looking at microRNA regulation? Was TonEBP found to undergo differential regulation between transcription and protein translation? The current transition in the story seems abrupt however the regulation of TonEBP and thermogenic genes is consistent with miR-30 control of TonEBP. Given that microRNA prediction software is notoriously promiscuous, the author's should address if the regulation of TonEBP 3' UTR is direct. Fusing the 3' UTR to a luciferase reporter is the gold standard in the microRNA field. Using the *Rip140* 3' UTR would be an excellent control given that is the target that the previous miR-30 beige fat paper proposed (PMID: 25576051). It will also be important to address how miR-30 control of TonEBP is related to miR-30 control of *RIP140* as both TonEBP and *RIP140* are transcriptional regulators?
2. The authors focus on the role of TonEBP in beiging of subcutaneous adipose almost exclusively. However, the earlier mechanisms proposed in adipocytes through general adipogenic pathways (*PPARg2*, mitotic clonal expansion, etc) would suggest more global effects in fat. What are the effects of TonEBP deletion in the main thermogenic adipose depot, brown fat? The gene expression effects in subQ in figure 2 are quite modest if compared to expression levels of many of those genes in brown fat (e.g. *Ucp1*). This should also be addressed in the adipose specific KO model (Figure 7).
3. The authors propose that TONEBP regulation of thermogenesis is through suppression of *Adrb3* expression and subsequent downstream targets. Yet cold exposure (through norepinephrine) will hit all the adrenergic receptors as well and many can compensate. What effect does TONEBP have on other beta-adrenergic receptors? The double knockdown experiment in figure 4E is a bit misleading to address *Adrb3*-dependence because knocking down *Adrb3* will dramatically effect cells by itself. To highlight this, the fold-increase in *Ucp1*, *Ppar-alpha*, and *Ppar-gamma* expression from TonEBP knockdown is almost the same with or without *Adrb3*. One of the best ways to address *ADRB3*-dependence would be through CL-316,243 (an *Adrb3* specific agonist) treatment of TonEBP KO mice or cells with or without TonEBP knockdown?
4. Given that TonEBP knockdown can increase differentiation the authors should attempt a cellular siRNA experiment in mature, differentiated adipocytes. From the materials and methods, it seemed that the siRNA studies were done in undifferentiated pre-adipocytes. Adipocytes that are simply more differentiated will yield similar findings and it's important to parse out the specific contribution of TonEBP to fat cell function.
5. The use of RG108 in wildtype mice makes the connection to TONEBP a far reach, particularly since treatment of cells did not affect TonEBP expression. In order to test the functional role of *Dnmt1* specifically in TonEBP control, TonEBP KO animals should be incorporated in the studies or cells with TonEBP siRNA knockdown should be investigated.
6. I commend the authors on the use of the conditional loss-of-function model. However, some metric of glycemic control test (GTT or ITT) would be important to show that the effects of whole body haplo-deficiency originated from adipose.
7. Does the beiging phenotype in TonEBP deficient mice occur at thermoneutrality? A temperature in which adrenergic signaling is lowest and browning does not occur physiologically.

Minor Points:

1. Are the TonEBP expression data in Figure 1A all set relative 1 in each control tissue? It's a more misleading presentation given Figure S1A showing just how poorly enriched TonEBP is in iWAT. This also gives concern for the use of a whole body knockout model in most of the paper.

2. The authors should also cite the following paper from Lee et al on "TonEBP/TONEBP inhibits adipocyte differentiation via modulation of mitotic clonal expansion during early phase of differentiation in 3T3-L1 cells."
3. Source data should be provided in a supplemental table for how the indirect calorimetry bar graphs next to each raw data trace were calculated (e.g. Figure 6B-D).
4. Were the cold challenge experiments that yielded gene expression (e.g. Figure 6F-G) done at the same time or with separate cohorts? As a positive control it's always nice to graph room temperature versus cold with and with genetic manipulation on the same graph. It's difficult to tell from these separated graphs if there is an interaction between TonEBP deficiency and cold (i.e. Adrb3 is induced equally in both 6F and 6G).
5. I wouldn't call leptin a "pro-diabetic adipokine"

Reviewers' comments:

Reviewer #1 (Remarks to the Author):

In the present study entitled 'TonEBP//NNFAT5 promotes obesity and insulin resistance by epigenetic suppression of white adipose tissue beiging', Lee et al. identify TonEBP as a repressor for some thermogenic genes.

As I am no expert in the physiology of fat tissue and obesity, I will comment only on the epigenetic aspect of this story. In general, the idea that TonEBP might regulate gene expression of thermogenic genes by recruitment of DNMT1 and subsequent DNA methylation seems straightforward. However, the model for this epigenetic regulation is still underdeveloped and require additional experimentation.

Specific points:

The study tries to link TonEBP-mediated recruitment of DNMT1 as a silencing mechanism of Adrp3. The model presented in Figure 6H proposes that this is a general mechanism to silence thermogenic/beiging genes. Given that the authors propose that TonEBP might be a potential therapeutic target it seems reasonable to probe this on a genome-wide scale. One way to address this would be to perform RNA-seq and compare the overlap of misregulated genes in TonEBP and DNMT1 knock-downs (knock-outs?).

-> We added PCR array data for 76 adipocyte physiology-related genes in 3T3-L1 cells after separate knockdown of TonEBP and DNMT1 (Figure S28). The results shows that 51 out of the 76 genes (67%) display concordance in the changes (up, down, or no change) of expression after knockdown of TonEBP vs. DNMT1. Thus, more than 60% of the 76 genes are likely to be regulated by the TonEBP-DNMT1 pathway potentially affecting many aspects of adipocyte physiology in addition to thermogenesis/beiging. We added a paragraph in Discussion (2nd paragraph, p15) to convey this new information.

Figure 4

The ChIPs should be compared to an independent locus to probe for differences in the IP. What is the standard deviation of the ChIP? There should at least be 3 independent, biological replicates!

-> We agreed and performed ChIP in other regions (Figure S24B). The results show that ChIP signal for the D region is not suppressed by TonEBP knockdown while the signal is decreased in the A, B, and C regions. We corrected n = 4 and added standard deviation for the ChIP data in Figure 4.

The effect on DNA methylation in the absence of TonEBP is very modest, particularly at positions A (TonEBP binding site) and C. This raises the question of how TonEBP recruits DNMT1 1.8kB further down at the core promoter (position B) to change/maintain DNA methylation? Generally, the TonEBP binding site is fairly far away (1.8kb) to be part of a promoter region (although not impossible). The question arises if this region is more of a proximal poised enhancer than promoter. In this case, TonEBP might loop to the

core promoter element. This might explain the more prominent effect on the core promoter. Experiments to be done should address if TonEBP is also recruited closer to TSS. The epigenetic landscape of this genomic region should be characterised in more detail, including ChIPs for H3K4me3, H3K27me3, H3K27ac and H3K4me1.

-> We agreed and added ChIP data for TonEBP and DNMT1 binding to the TSS region (Figure S24 A and B). In addition, epigenetic changes (H3K27me3, H3K4me3 and H3Ac) relevant to transcriptional regulation were examined (Figure 4K and Figure S20). The additional data now show that B region, which is ~1.5 kB downstream of the TonEBP binding site, has higher level of TonEBP-dependent TonEBP-DNMT1 recruitment compared to A, C, or D region. As expected, epigenetic status of B region favors TonEBP-dependent transcriptional suppression due to reduced H3K4 tri-methylation and H3 acetylation in combination with elevated H3K27 tri-methylation. We added a paragraph in p11 and two sentences in p12 to describe the new information.

Additionally, given the modest effects of DNA methylation in the absence of TonEBP, is forced recruitment of DNMT1 (e.g. by inactive Cas9) in the absence of TonEBP sufficient to keep *Adrb3* expression low?

-> We did not attempt forced recruitment to the *Adrb3* promoter. However, it is known that DNMT1-induced DNA methylation can lead to histone modifications (34 – 36). Indeed, our data show DNMT1-induced changes in histone methylation and acetylation by DNMT1 which favor transcriptional suppression (see above). These results can explain how modest changes in DNA methylation regulate *Adrb3* expression.

In contrast to the high methylation rate in NIH3T3-L1s (Figure 4), the methylation levels in vivo are relatively low (around 30% - Figure S12B) – is there an additional pathway regulating *Adrb3*? The authors should at least comment on this point!

-> 3T3-L1 cells have more fibroblast-like characteristics than adipocyte. The expression of *Adrb3*, an adipocyte-specific gene, is lower in 3T3-L1 cells compared to primary adipocytes. The high DNA methylation rate of the *Adrb3* promoter in 3T3-L1 cells is consistent with the low expression of *Adrb3*. Our data show that DNA methylation of the *Adrb3* promoter is lower in iWAT and primary adipocytes (Figure S22A and B) compared to 3T3-L1 cells (Figure 4L).

We added this comment on manuscript (page 11):

“We found that basal levels of DNA methylation were much higher in 3T3-L1 cells compared to iWAT and primary adipocytes (**Figure 4L, Figure S22A and B**). Consistent with this, we also found that *Adrb3* mRNA expression is 50 – 100 times higher in iWAT and primary adipocytes (data not shown) indicating that 3T3-L1 cells have much more of fibroblastic characteristics.”

General points:

miRNA 30 is introduced very rapidly – Has there been a link with TonEBP before? It should maybe be introduced shortly in the introduction.

-> Yes, miRNAs have been linked to TonEBP expression. We added the following text in Introduction (page 4):

“MicroRNAs (miRs), short noncoding RNAs, block translation or induce degradation of their target mRNAs by base-pairing to the recognition sites (19). Recently, the associations between TonEBP and miRs have been reported in cancer and diabetes (20, 21, 22), but there has been no report in adipose tissues.”

Please always indicate in the legends what cell types have been used for the experiments, as different systems are used for different experiments and in the present form it is sometimes difficult to understand whether primary tissues/cells or cell lines were used. I suppose that it's a heterozygous knock-out mouse that was used in Fig S12B?

-> Yes, it was from heterozygotes (Figure S22A in the revised manuscript). We revised all the relevant legends to indicate genotypes, and whether primary tissues/cells or cell lines were used for the experiments.

Reviewer #2 (Remarks to the Author):

In this work Tonicity-responsive Enhancer Binding Protein (TonEBP) is presented as an epigenetic regulator leading to obesity and insulin resistance. Its expression is upregulated in murine and human models of obesity, resulting in suppression of thermogenic capacity and of browning in subcutaneous white adipose tissue. TonEBP role in the context of obesity was investigated using TonEBP haplo-deficient mice fed chow or high fat diet and TonEBP knock-down cultured adipocytes. Authors demonstrate that decreased TonEBP expression leads to reduced body weight, increased energy expenditure and thermogenesis, ameliorated glucose metabolism and insulin resistance. TonEBP increases DNA methylation on the promoter of beta-3 adrenoreceptor gene (*Adrb3*), by recruiting DNMT1 DNA methylase, thereby negatively regulating thermogenic and browning gene expression. Furthermore, a link between TonEBP and miR30 is reported.

Based on the results showed, TonEBP appears as an interesting and novel target linking adipose tissue metabolism, inflammation and systemic alterations in obesity and epigenetic regulation.

I have the following concerns regarding this manuscript:

Main concerns:

1. 3T3-L1 knock-down adipocytes do not completely recapitulate the in vivo effects. For example, UCP1 expression is increased in iWAT and SVF from haplo-deficient mice while this is not observed in 3T3-L1 knocked-down cells. Moreover, the expression of some of the measured genes is not strongly increased, raising some doubts about biological relevance of these differences. Figure 4E shows 20-fold induction of *Cidea* while this is not observed in Figure S6A, even though experimental conditions are similar (Con-Scr siRNA vs Con-TonEBP siRNA in Fig S6A and Scr siRNA vs TonEBP siRNA in Fig. 4E). For the above-mentioned reasons, I doubt this in vitro model could be the more appropriate to study TonEBP regulation. Authors should consider repeating some experiments in another adipocyte cell line. In particular, they should verify experiments in Figure S6A and 4E. Similarly, it would be greatly appreciable if authors could confirm TonEBP ChIP results shown in Fig 4H either in an alternative adipocyte cell line or in vivo in iWAT.

-> We agreed and confirmed all the data obtained from 3T3-L1 cells using iWAT and primary adipocytes (Figure 2J, 2K, S9, S10, S18B and S18C).

2. TonEBP is overexpressed to evaluate its impact on thermogenic gene expression (Figure S6B). However, TonEBP mRNA levels are not significantly increased, thereby no conclusive explanation on observed suppression of thermogenic genes is possible. Since mRNA levels did not increase much upon overexpression of TonEBP, authors should check whether Adenovectors increased TonEBP protein levels.

-> We added immunoblotting of TonEBP which confirmed the over-expression (Figure S11B).

3. Authors analysed all data using Student's t-test. However, if groups are more than two, Student's t-test is not the most appropriate and suitable statistical test. Standard deviation is missing in the Figures 4H, K, L, M, S12B. Moreover, authors did not report any statistical significance in Figures 4H, K, L, M.

-> We performed 1-way ANOVA in Figures 4E, S3B and C, S9 and Figure S11. We revised statistical analyses in Figure legends and Methods accordingly. Also, we added standard deviation and statistical significance (Figures 4H, K, L, M and S22A).

4. Authors did not rule out that increased thermogenesis could be due to possible effects of the haplo-deficiency of TonEBP in BAT. They should measure gene expression in BAT to provide information about BAT contribution on the observed phenotype.

-> TonEBP expression in BAT was not induced in response to HFD (Figure 1A). Also, we checked UCP-1 expression in iWAT, eWAT and BAT of WT and TonEBP haplo-deficient mice and results showed no significant change of UCP-1 expression in BAT (Figure S8). These data indicate that TonEBP regulates thermogenesis in iWAT, but not in BAT. We revised text in line 6 of p7 accordingly.

5. Basal temperature and cold exposure were carried out in CD-fed mice (Figure 2D and E). Considering the role of TonEBP in response to HFD, it would be a stronger evidence to perform these experiments also in HFD setting.

-> We agreed and added basal and cold exposure rectal temperature data on HFD fed animals (Figure S7) and adipocyte-specific TonEBP KO mice (Figure S27A).

6. The WB shown in Figure 1B does not include a reference control protein to verify equal loading. Could you please add it?

-> We added Hsc70 (house-keeping protein).

Minor concerns:

1. Quantification is missing in all western blots. WB quantification by densitometry should be reported.

-> We added graphs on WB quantification (Figure 1B and 1F, Figure 2H, and Figure 3D).

2. Figure 1A: Due to high expression of TonEBP in iWAT in mice on HFD, the scale does not allow to appreciate the expression in other tissues. Therefore, I recommend to show data from iWAT and from other tissues separately in different panels to better emphasize differences also in other tissues.

-> We agreed and separated iWAT data from other tissues (Figure 1A).

3. Figure 1E: The WB presents two samples per group while n=4 is reported in the caption. Could you clarify this point?

-> We moved '(n=4)' to Figure 1D.

4. Figure 1I: In the sentence at page 6 “Examination of dissected fat tissues confirmed that epididymal, inguinal, and dorsal fat pads were smaller in these animals” the word “fat” is misspelled (“fad” instead of “fat”) and should be corrected. Furthermore, this figure shows the reduction of adipose tissue mass while Figure S4 reports the absence of alterations in food intake. In the first line of the paragraph “Adipocyte TonEBP suppresses thermogenesis and beiging of WAT” (page 6, line 16), the reference to Figure S4 where you mention food intake in the text should be added to the reference to Figure 1I.

-> We corrected fad to fat, and Figure 4S was added.

5. Figure 2A,B, 6B,C: Authors should calculate and report the Respiratory Exchange Ratio.

-> We added RER for TonEBP haplo-deficient mice (Figure S6) and adipocyte specific TonEBP KO mice (Figure S26). Texts were revised accordingly in p13.

6. Figure 2D and E: The basal body temperature is different between the two figures. Furthermore, at time 0 of cold exposure the difference between the two groups is not statistically significant whereas it is in Figure 2D. How would you explain this discrepancy?

-> In order to perform the cold exposure experiments, mice were moved from the animal facility to a laboratory equipped with cold room. This environmental change might have affected the animals' basal temperature. We added body temperature data from HFD-fed animals (Figure S7) which show higher basal as well as cold exposure body temperature. We believe the basal body temperature is higher in the heterozygotes.

7. Figure 2H and I: The diet used in this experiment should be reported.

-> We added CD to the legend.

8. Figure 2J and K: At pg. 7, authors stated in the text “In adipocytes differentiated from SVF of the TonEBP haplo-deficient animals or 3T3-L1 cells whose TonEBP was knocked down showed elevated basal and isoproterenol-stimulated expression of thermogenic genes...”. This sentence is ambiguous as it sounds like both adipocytes from SVF and 3T3-L1 adipocytes were treated with isoproterenol, while this does not seem the case (i.e., adipocytes from SVF were not treated with isoproterenol, while 3T3-L1 adipocytes were treated with isoproterenol).

-> We agreed and revised manuscript (p 7):

“Next, we examined adipocytes *in vitro*. Adipocytes differentiated from stromal vascular fractions of the TonEBP haplo-deficient animals showed elevated expression of thermogenic genes, hormone-sensitive lipase (*HSL*) and beige marker genes (**Figures 2J, K and S10**). In addition, TonEBP knockdown (**Figures S11A**) enhanced and adenovirus-mediated overexpression of TonEBP after differentiation reduced the basal and isoproterenol-stimulated expression of these genes in differentiated 3T3-L1 cells (**Figure S11B**).”

9. Figure 3D, E, F: Authors should provide a rationale whereby the analyses in these panels were performed in eWAT and not iWAT. Is this because eWAT is more relevant in potentially affecting insulin signalling and adipokine secretion owing to inflammation induced by HFD?

-> Yes. Insulin resistance and changes in adipokine secretion were known to occur mainly in eWAT due to HFD-induced local inflammation (26). We revised manuscript to add this rationale (p 8).

10. Figure 3G: The bar scale should be reported indicating the unit.

-> We added bar scale in the legend.

11. Figure 3N: Could you uniform the unit g/ L of total cholesterol to mg/dL?

-> We revised the unit to mg/dL.

12. Figure 4H, K, M: Authors should report enrichment in a DNA negative control region to show that enrichment is specific for the considered chromatin region.

-> We agreed and added a negative region (D) (Figure S24A and B).

13. Figure 4L: This panel is not clear at all. Please, provide a clear explanation of the figure either in the text or in the figure legend.

-> We added Figure S21 for methylation sites and sequences for A, B, and C, and explanation was added to legend. Experimental procedure is detailed in Materials and Methods.

14. Materials and methods: Mice. Please specify the mice sub-strain (C57BL/6J or C57BL/6N?). Cells and reagents: Authors stated that they used RAW264.7 cells, but no data with this cell line are shown. Chromatin Immunoprecipitation assay: Authors stated that cells were grown with LPS. This is probably a typing error.

-> It is C57BL/6J and text was revised. We removed typing error: RAW264.7 cells and LPS.

15. Figure S1B: TonEBP mRNA expression was elevated in iWAT from 16-week-old db/db obese mice compared to their lean db/+ littermates (page 5). In the caption is reported that mice were 10-week-old. The age of these mice should be reported consistently in the text and in figure caption.

-> We revised manuscript in p5 to 10-week.

16. Figure S13: These correlations are not totally conclusive. Authors should tone down the conclusion in the text stating "These results demonstrate that TonEBP promotes DNA methylation of the Adrb3 promoter in adipocytes of mice and humans". The graph in Figure S13A only shows a negative correlation between DNA methylation of the ADRB3 promoter and the levels of ADRB3 mRNA in human subWAT. Nonetheless, this does not represent the proof that TonEBP promotes DNA methylation of ADRB3

promoter in humans: it's only a correlation that could suggest a possible role of TonEBP in human WAT physiology speculating on the observations in TonEBP^{+/-} mice. Also, note that human gene symbols should be upper case (i.e., ADRB3, not Adrb3).

-> We revised the text accordingly (p11 - 12) and human gene symbols:

We found that in human subcutaneous adipocytes DNA methylation of the *ADRB3* promoter correlated negatively with *ADRB3* mRNA expression (**Figure S23A**), while correlated positively with BMI (**Figure S23B**). These data suggest that TonEBP also promotes DNA methylation of the *ADRB3* promoter in human adipocytes in view of the inverse relationship between mRNA levels of *ADRB3* vs. *TONEBP* (**Figure 4A**).

Reviewer #3 (Remarks to the Author):

In this article, Lee et al. seek to investigate the role of transcriptional regulator TonEBP in beiging of adipose tissue and the functional consequences on systemic metabolism. The authors used both whole body and adipose tissue specific KOs as well as pharmacological compounds to tackle various aspects of adipose metabolism and TonEBP biology. While TonEBP-regulation of the *Adrb3* promoter through *Dnmt1* was shown convincing, serious weaknesses exist in causally linking that single event to the systemic effects observed in the animals. Furthermore, as TonEBP has been previously shown to transcriptionally control adipocyte differentiation and subsequently insulin sensitivity, I'm concerned about the significance of the conceptual advance offered by this work in its current form. In order to be appropriate for publication in *Nature Communications*, I believe the following points should be addressed.

Major Points:

1. What was the rationale for looking at microRNA regulation? Was TonEBP found to undergo differential regulation between transcription and protein translation? The current transition in the story seems abrupt however the regulation of TonEBP and thermogenic genes is consistent with miR-30 control of TonEBP. Given that microRNA prediction software is notoriously promiscuous, the author's should address if the regulation of TonEBP 3' UTR is direct. Fusing the 3' UTR to a luciferase reporter is the gold standard in the microRNA field. Using the *Rip140* 3' UTR would be an excellent control given that is the target that the previous miR-30 beige fat paper proposed (PMID: 25576051). It will also be important to address how miR-30 control of TonEBP is related to miR-30 control of *RIP140* as both TonEBP and *RIP140* are transcriptional regulators?

-> We performed luciferase reporter assay to demonstrate direct binding of miR-30 to 3' UTR of TonEBP (Figure S3C). We find miR-30 binds to two sites in TonEBP 3' UTR and suppresses TonEBP expression like *RIP140* expression. In addition we added the following sentence in p6 to discuss *RIP140*: "Thus, miR-30 promotes thermogenesis and beiging *via* suppression of two transcriptional regulators – TonEBP and *RIP140* (23)."

2. The authors focus on the role of TonEBP in beiging of subcutaneous adipose almost exclusively. However, the earlier mechanisms proposed in adipocytes through general adipogenic pathways (*PPARg2*, mitotic clonal expansion, etc) would suggest more global effects in fat. What are the effects of TonEBP deletion in the main thermogenic adipose depot, brown fat? The gene expression effects in subQ in figure 2 are quite modest if compared to expression levels of many of those genes in brown fat (e.g. *Ucp1*). This should also be addressed in the adipose specific KO model (Figure 7).

-> Yes, we previously reported that TonEBP suppresses adipogenic pathway *in vitro*. Here we find that TonEBP haplo-deficient animals display more changes related with thermogenesis (enhanced energy expenditure, enhanced body temperature, reduced fat mass, enhanced thermogenic gene expression and etc) rather than adipogenesis (no effect on *PPARg* expression, Figure 2). This was why we focused on the beiging pathway. Since *TonEBP* mRNA was not affected by HFD (Figure 1A) and *Ucp-1* mRNA expression was not affected by TonEBP deficiency (Figure S8) in brown fat, we did not explore gene expression profile in brown fat of the animals with adipocyte-specific

TonEBP deletion.

3. The authors propose that TONEBP regulation of thermogenesis is through suppression of *Adrb3* expression and subsequent downstream targets. Yet cold exposure (through norepinephrine) will hit all the adrenergic receptors as well and many can compensate. What effect does TONEBP have on other beta-adrenergic receptors? The double knockdown experiment in figure 4E is a bit misleading to address *Adrb3*-dependence because knocking down *Adrb3* will dramatically effect cells by itself. To highlight this, the fold-increase in *Ucp1*, *Ppar-alpha*, and *Ppar-gamma* expression from TonEBP knockdown is almost the same with or without *Adrb3*. One of the best ways to address ADRB3-dependence would be through CL-316,243 (an *Adrb3* specific agonist) treatment of TonEBP KO mice or cells with or without TonEBP knockdown?

-> We agreed and performed CL-316,243 treatment to primary adipocytes from WT and TonEBP haplo-deficient mice. We found that cAMP production in response to CL-316,243 treatment was higher in TonEBP haplo-deficient cells (Figure S16) consistent with higher expression of ADRB3. *Ucp-1* mRNA expression was elevated both in WT and TonEBP haplo-deficient adipocytes (Figure S17). The ~2-fold higher *Ucp-1* mRNA expression in TonEBP haplo-deficient adipocytes after CL-316,243 treatment was comparable to ~2-fold higher *Ucp-1* mRNA expression in TonEBP haplo-deficient iWAT after cold exposure (Figure S9). These data are consistent with the idea that TonEBP regulation of thermogenesis is through suppression ADRB3 expression. Texts were added to p10 to describe these new data.

4. Given that TonEBP knockdown can increase differentiation the authors should attempt a cellular siRNA experiment in mature, differentiated adipocytes. From the materials and methods, it seemed that the siRNA studies were done in undifferentiated pre-adipocytes. Adipocytes that are simply more differentiated will yield similar findings and it's important to parse out the specific contribution of TonEBP to fat cell function.

-> We performed adenovirus-mediated TonEBP overexpression on fully differentiated 3T3-L1 cells (Figure S11B). This results show that TonEBP suppresses thermogenic genes in the differentiated adipocytes. Texts were added to p7 to incorporate these new data.

5. The use of RG108 in wildtype mice makes the connection to TONEBP a far reach, particularly since treatment of cells did not affect TonEBP expression. In order to test the functional role of *Dnmt1* specifically in TonEBP control, TonEBP KO animals should be incorporated in the studies or cells with TonEBP siRNA knockdown should be investigated.

-> We treated RG108 to examine whether DNMT inhibition in cell and mice has similar thermogenic phenotypes with TonEBP deficiency. We performed siTonEBP/siDNMT1 double knock-down experiment in 3T3-L1 adipocytes. The results show that TonEBP/DNMT1 double knock-down did not lead to significant changes in *Ucp-1* expression compared to single knock-downs (Figure S25). These data suggest that TonEBP and DNMT1 are in the same pathway. Text was added to p13 to describe this

new information.

6. I commend the authors on the use of the conditional loss-of-function model. However, some metric of glycemic control test (GTT or ITT) would be important to show that the effects of whole body haplo-deficiency originated from adipose.

-> We agreed and added GTT and ITT in adipocyte-specific TonEBP KO mice (Figure S27B). The results are essentially the same with the TonEBP haplo-deficient animals: higher body temperature (thermogenesis) and better insulin sensitivity. Texts in p13 were revised to incorporate these new data.

7. Does the beiging phenotype in TonEBP deficient mice occur at thermoneutrality? A temperature in which adrenergic signaling is lowest and browning does not occur physiologically.

-> Although we could not perform thermo neutrality experiments, we imagine that TonEBP-deficient animals would not display beiging phenotype when they are fed CD and kept under thermoneutral conditions. This is because expression of *Adrb3* (Figure 4B) and thermogenic genes (Figure S9) is unchanged when these animals are fed CD at room temperature. On the other hand, we suspect that HFD feeding could induce thermogenic phenotype in these animals even under thermoneutral conditions because HFD feeding at room temperature leads to enhanced expression of *Adrb3* (Figure 4B) and thermogenic genes (Figure 2F, G).

Minor Points:

1. Are the TonEBP expression data in Figure 1A all set relative 1 in each control tissue? It's a more misleading presentation given Figure S1A showing just how poorly enriched TonEBP is in iWAT. This also gives concern for the use of a whole body knockout model in most of the paper.

-> Yes, Figure 1A data are relative 1 to each control tissue to show TonEBP induction in response to HFD. We agreed and removed Figure S1A. Immunoblot signal of TonEBP was clear and strong (Figure 1B).

2. The authors should also cite the following paper from Lee et al on "TonEBP/TONEBP inhibits adipocyte differentiation via modulation of mitotic clonal expansion during early phase of differentiation in 3T3-L1 cells."

-> We agreed and cited this paper (18) in p4.

3. Source data should be provided in a supplemental table for how the indirect calorimetry bar graphs next to each raw data trace were calculated (e.g. Figure 6B-D).

-> Since the excel files containing raw data for Figure 2A-C and Figure 6B-D are too big for Supplementary table, we added data at the end of the files for your examination.

4. Were the cold challenge experiments that yielded gene expression (e.g. Figure 6F-G)

done at the same time or with separate cohorts? As a positive control it's always nice to graph room temperature versus cold with and with genetic manipulation on the same graph. It's difficult to tell from these separated graphs if there is an interaction between TonEBP deficiency and cold (i.e. Adrb3 is induced equally in both 6F and 6G).

-> Figure 6F and 6G are obtained from separate cohorts. Although we do not have simultaneous HFD and cold exposure data in adipocyte-specific TonEBP KO animals, we do have them in TonEBP haplo-deficient mice and added the data in Figure S9.

5. I wouldn't call leptin a "pro-diabetic adipokine"

-> We agreed and removed this phrase.

Date	VCC										VCC										VCC										VCC									
	W1	W2	W3	W4	W5	W6	W7	W8	W9	W10	W1	W2	W3	W4	W5	W6	W7	W8	W9	W10	W1	W2	W3	W4	W5	W6	W7	W8	W9	W10	W1	W2	W3	W4	W5	W6	W7	W8	W9	W10
1	0.000000	0.000000	0.000000	0.000000	0.000000	0.000000	0.000000	0.000000	0.000000	0.000000	0.000000	0.000000	0.000000	0.000000	0.000000	0.000000	0.000000	0.000000	0.000000	0.000000	0.000000	0.000000	0.000000	0.000000	0.000000	0.000000	0.000000	0.000000	0.000000	0.000000	0.000000	0.000000	0.000000	0.000000	0.000000	0.000000	0.000000	0.000000	0.000000	0.000000

W1	W2	W3	W4	W5	W6	W7	W8	W9	W10
0.000000	0.000000	0.000000	0.000000	0.000000	0.000000	0.000000	0.000000	0.000000	0.000000

Reviewers' comments:

Reviewer #1 (Remarks to the Author):

In the revised version of Lee et al. 'TonEBP//NNNFAT5 promotes obesity and insulin resistance by epigenetic suppression of white adipose tissue beiging', the authors have addressed almost all of my concerns. The authors have performed additional experiments addressing the epigenetic landscape. Unfortunately, two marks are still missing that would help to define regions A and B better and help to distinguish enhancer and promoter of *Adrb3*: H3K27ac and H3K4me1. As the authors propose a looping model, which agrees with the data presented so far, I feel that it is important to address this point – the experiment should also be easily done in a cell line. All the marks tested so far just underscore the transcriptional state of *Adrb3*. Finally, it would be interesting to analyze the promoters/enhancers of genes tested in Figure S28 and identify potential TonEBP binding motifs to understand if these genes might be direct targets of TonEBP.

Minor points:

Supplementary Figures should be combined in a logical fashion with respect to the main figures in the text to facilitate reading.

Fig. 1G: The authors should indicate time point of photo (is it 14 weeks?).

S18: Please add promoter scheme to facilitate understanding. In this case one does not have to go back to main figure.

S24: Define # and normalize the ChIPs to input.

Reviewer #2 (Remarks to the Author):

The authors made significant effort to improve the manuscript according to the major and minor concerns raised by the reviewers. There are still some points that need to be clarified/reviewed before the manuscript could be accepted. Here are the points that need some attention:

1. There are discrepancies in gene expression between SVCs differentiated to adipocytes in Fig. 2J and 2K vs results in Fig. S9: why in Fig. 2J and K many genes are upregulated in aplodeficient SVC-derived adipocytes while in iWAT of TonEBP aplodeficient mice they are no? (see *Ucp1*, *Pgc1*, *Dio2*, *Cidea*, *Ppara*, *Pparg*, *Hsl* that are upregulated in SVCs from TonEBP aplodeficient mice, while the same genes are not upregulated in iWAT of aplodeficient TonEBP mice at room temperature).

2. At pg. 12 last four lines, the authors stated "The same interaction was also observed in the B and C regions suggesting that TonEBP-DNMT1 complex bound to the A region made contact with the B and C regions via looping." However, the binding of the TonEBP-DNMT1 complex does not necessarily imply DNA looping and may make contacts with several regions of the genome (A, B, C) even without looping. I suggest to simply state "The same interaction was also observed in the B and C regions suggesting that TonEBP-DNMT1 complex bound to the A region made also contacts with the B and C regions."

Also, in the legend of Fig. S24 the authors should specify that region D is a DNA negative control region.

3. At pg. 12 lines 6-10, the authors corrected the previous statement with "We found that in human subcutaneous adipocytes DNA methylation of the *ADRB3* promoter correlated negatively with *ADRB3* mRNA expression (Figure S23A), while correlated positively with BMI (Figure S23B). These data suggest that TonEBP also promotes DNA methylation of the *ADRB3* promoter in human adipocytes in view of the inverse relationship between mRNA levels of *ADRB3* vs. TONEBP (Figure 4A)." I recommend to further modify this statement as follows: "We found that in human subcutaneous adipocytes DNA methylation of the *ADRB3* promoter correlated negatively with *ADRB3* mRNA expression (Figure S23A), while correlated positively with BMI (Figure S23B). These data suggest that TonEBP may promote DNA methylation of the *ADRB3* promoter also in human adipocytes in view of the inverse relationship between mRNA levels of *ADRB3* vs. TONEBP (Figure

4A)."

Reviewer #3 (Remarks to the Author):

I commend the authors on addressing my concerns experimentally however, I'm still uncertain about the level of the conceptual leap in terms of appropriateness for Nature Communications. Ultimately, I feel the studies are at least soundly executed enough to warrant consideration.

Reviewers' comments:

Reviewer #1 (Remarks to the Author):

In the revised version of Lee et al. 'TonEBP//NNNFAT5 promotes obesity and insulin resistance by epigenetic suppression of white adipose tissue beiging', the authors have addressed almost all of my concerns. The authors have performed additional experiments addressing the epigenetic landscape.

Unfortunately, two marks are still missing that would help to define regions A and B better and help to distinguish enhancer and promoter of *Adrb3*: H3K27ac and H3K4me1. As the authors propose a looping model, which agrees with the data presented so far, I feel that it is important to address this point – the experiment should also be easily done in a cell line. All the marks tested so far just underscore the transcriptional state of *Adrb3*.

We agreed and newly examined ChIPs for H3K4me1, H3K4me3, H3K27me3 and H3K27ac on A and B regions. The new data show that monomethylation (H3K4me1) and trimethylation of H3K4 (H3K4me3), and acetylation of H3K27 (H3K27ac) were elevated while trimethylation of H3K27 (H3K27me3) was reduced by TonEBP knockdown at both the A and B regions (Fig. 4k). In addition, we carried out ChIP analysis on the *GAPDH* promoter. The epigenetic marks in the *GAPDH* promoter were not affected by TonEBP knockdown (supplementary figure 5h in the revised manuscript). We revised the paragraph in p11 and corresponding figure legends and methods to incorporate the new information.

Finally, it would be interesting to analyze the promoters/enhancers of genes tested in Figure S28 and identify potential TonEBP binding motifs to understand if these genes might be direct targets of TonEBP.

The consensus motif of TonEBP binding site is TGGAAANNYNY. We searched this motif on the promoters/enhancers (defined as 0 – 5Kb from TSS) of top 20 upregulated and top 5 downregulated genes in response to TonEBP knockdown listed in Figure S28 (supplementary figure S11 in the revised manuscript) (see table below). In 17 of these genes, at least one TonEBP binding motif is located in the promoter/enhancer region while none is found in 5 genes.

No.	Gene	TonEBP knockdown-mediated change	Position in the genome	RefSeq	Location of TonEBP binding site
1	cidea	UP	Chromosome [NC_000084.6]; Strand [+]; Position [67343479]	NM_007702	-4701 to -4694 (TGGAAATTTTT)
2	slc2a4 (glut4)	UP	Chromosome [NC_000077.6]; Strand [-]; Position [69948161]	NM_009204	none
3	cebpb	UP	Chromosome [NC_000068.7]; Strand [+]; Position [167688967]	NM_009883	-3055 to -3065 (TGGAAATGCCT)
4	Ddit3	UP	Chromosome [NC_000076.6]; Strand [+]; Position [127290797]	NM_007837	none
5	Agt	UP	Chromosome [NC_000074.6]; Strand [-]; Position [124569706]	NM_007428	-2262 to -2272 (TGGAAAATGT); -2435 to -2445 (TGGAAAGACAC); -3910 to -3920 (TGGAAACTCC);
6	adrb3	UP	Chromosome [NC_000074.6]; Strand [+]; Position [129401213]	NM_013462	-1808 to -1818 (TGGAAAATTT)
7	ucp1	UP	Chromosome [NC_000074.6]; Strand [+]; Position [83290352]	NM_009463	none
8	Egr2	UP	Chromosome [NC_000076.6]; Strand [+]; Position [67537870]	NM_010118	-3580 to -3590 (TGGAAACGCAT)
9	Pgc1b	UP	Chromosome [NC_000084.6]; Strand [-]; Position [61400400]	NM_133249	none
10	Jun	UP	Chromosome [NC_000070.6]; Strand [-]; Position [95052196]	NM_010591	none
11	Sirt3	UP	Chromosome [NC_000073.6]; Strand [-]; Position [140881825]	NM_022433	-979 to -989 (TGGAAAGGTTC)
12	fabp4	UP	Chromosome [NC_000069.6]; Strand [-]; Position [10208576]	NM_024406	-1485 to -1495 (TGGAAACGTAT); -4758 to -4768 (TGGAAACCCAC)
13	Cfd	UP	Chromosome [NC_000076.6]; Strand [+]; Position [79890910]	NM_013459	-3532 to -3542 (TGGAAAGGCAC)
14	ppara	UP	Chromosome [NC_000081.6]; Strand [+]; Position [85735526]	NM_011144	-2149 to -2159 (TGGAAAGGTCT)

15	Gata3	UP	Chromosome [NC_000084.6]; Strand [+]; Position [67343479]	NM_007702	none
16	Cdkn1a	UP	Chromosome [NC_000083.6]; Strand [+]; Position [29093772]	NM_007669	none
17	Adig	UP	Chromosome [NC_000068.7]; Strand [+]; Position [158502543]	NM_145635	-6 to -16 (TGGAAAGGCC); -2596 to -2606 (TGGAAAACTT) -2981 to -2991 (TGGAAATATCC)
18	pparg	UP	Chromosome [NC_000072.6]; Strand [+]; Position [115361269]	NM_011146	-1274 to -1284 (TGGAAATTTAT) -3948 to -3958 (TGGAAATTCT)
19	Ppard	UP	Chromosome [NC_000083.6]; Strand [+]; Position [28232741]	NM_011145	-1410 to -1420 (TGGAAACCTAT)
20	Foxo1	UP	Chromosome [NC_000069.6]; Strand [+]; Position [52268418]	NM_019739	none
21	Ccnd1	Down	Chromosome [NC_000073.6]; Strand [-]; Position [144939831]	NM_007631	-1238 to -1248 (TGGAAACGTTT)
22	Angpt2	Down	Chromosome [NC_000074.6]; Strand [-]; Position [18741548]	NM_007426	-442 to -452 (TGGAAAAGCAC) -3304 to -3314 (TGGAAATGTCT)
23	Fgf2	Down	Chromosome [NC_000069.6]; Strand [+]; Position [37348667]	NM_008006	-247 to -257 (TGGAAATCTCC)
24	Gata2	Down	Chromosome [NC_000072.6]; Strand [+]; Position [88198668]	NM_008090	-1257 to -1267 (TGGAAAGCCCT); -3575 to -3585 (TGGAAATCCAC) -4220 to -4230 (TGGAAAGACT)
25	Runx1t1	Down	Chromosome [NC_000070.6]; Strand [+]; Position [13743360]	NM_009822	-858 to -868 (TGGAAATATCT); -3997 to -4007 (TGGAAATGCCT) -4051 to -4061 (TGGAAATTCT); -4411 to -4421 (TGGAAACCTGC)

TonEBP is recruited to the promoter/enhancer regions of its target genes in two different ways: by DNA binding to its binding motif (5, 9, this study) or by interactions with other proteins without DNA binding (10, 11, 17). Thus, DNA binding is not required for gene regulation by TonEBP as discussed in the second paragraph in p16; the results described here are consistent with this.

Minor points:

Supplementary Figures should be combined in a logical fashion with respect to the main figures in the text to facilitate reading.

: We combined the 28 supplementary figures into 11 figures in line with the main figures.

Fig. 1G: The authors should indicate time point of photo (is it 14 weeks?).

: We added 16 weeks in Fig. 1g.

S18: Please add promoter scheme to facilitate understanding. In this case one does not have to go back to main figure.

: We added promoter scheme (supplementary figure 6e in the revised manuscript) and revised figure legend.

S24: Define # and normalize the ChIPs to input.

: Data normalized to input and # defined (supplementary figure 9b and c in the revised manuscript). We revised figure legend accordingly.

Reviewer #2 (Remarks to the Author):

The authors made significant effort to improve the manuscript according to the major and minor concerns raised by the reviewers. There are still some points that need to be clarified/reviewed before the manuscript could be accepted. Here are the points that need some attention:

1. There are discrepancies in gene expression between SVCs differentiated to adipocytes in Fig. 2J and 2K vs results in Fig. S9: why in Fig. 2J and K many genes are upregulated in haplodeficient SVC-derived adipocytes while in iWAT of TonEBP haplodeficient mice they are no? (see Ucp1, Pgc1, Dio2, Cidea, Ppara, Pparg, Hsl that are upregulated in SVCs from TonEBP haplodeficient mice, while the same genes are not upregulated in iWAT of haplodeficient TonEBP mice at room temperature).

: The SVCs in Fig 2j and 2k are comparable to cold-exposure conditions Fig 2h and Supplementary Fig 3d because these cells were stimulated with thermogenic stimuli. Although iWAT of CD-fed TonEBP-deficient mice exhibit higher expression of UCP-1 ($p < 0.06$) and PGC1 α ($p < 0.055$) at room temperature, the differences are not statistically significant (Supplementary figure 3d). Most likely this reflects weak thermogenic signals at room temperature. On the other hand, when fed with HDF the differences become significant (Fig 2f and 2g) probably because the expression in wild type animals are reduced as a result of elevated TonEBP levels (Fig 1a and 1b).

2. At pg. 12 last four lines, the authors stated “The same interaction was also observed in the B and C regions suggesting that TonEBP-DNMT1 complex bound to the A region made contact with the B and C regions via looping.” However, the binding of the TonEBP-DNMT1 complex does not necessarily imply DNA looping and may make contacts with several regions of the genome (A, B, C) even without looping. I suggest to simply state “The same interaction was also observed in the B and C regions suggesting that TonEBP-DNMT1 complex bound to the A region made also contacts with the B and C regions.”

: We agreed and revised manuscript accordingly (p12 - 13).

Also, in the legend of Fig. S24 the authors should specify that region D is a DNA negative control region.

: We agreed and revised the legend of Fig. S24 manuscript (supplementary figure 9b and c).

“TonEBP and DNMT1 bind to the A, B and C regions but not D region, a negative control region”

3. At pg. 12 lines 6-10, the authors corrected the previous statement with “We found that in human subcutaneous adipocytes DNA methylation of the ADRB3 promoter correlated negatively with ADRB3 mRNA expression (Figure S23A), while correlated positively with BMI (Figure S23B). These data suggest that TonEBP also promotes DNA methylation of the ADRB3 promoter in human adipocytes in view of the inverse relationship between mRNA levels of ADRB3 vs. TONEBP (Figure 4A).” I recommend to further modify this statement as follows: “We found that in human subcutaneous adipocytes DNA methylation of the ADRB3 promoter correlated negatively with ADRB3 mRNA expression (Figure S23A), while correlated positively with BMI (Figure S23B). These data suggest that TonEBP may promote DNA methylation of the ADRB3 promoter also in human adipocytes in view of the inverse relationship between mRNA levels of ADRB3 vs. TONEBP (Figure 4A).”

: We agreed and revised manuscript (p 12).

“We found that in human subcutaneous adipocytes DNA methylation of the *Adrb3* promoter correlated negatively with *Adrb3* mRNA expression (Supplementary Figure 8d), while correlated positively with BMI (Supplementary Figure 8e). These data suggest that TonEBP may promote DNA methylation of the *Adrb3* promoter also in human adipocytes in view of the inverse relationship between mRNA levels of *Adrb3* vs. TonEBP (Fig. 4a).”

Reviewer #3 (Remarks to the Author):

I commend the authors on addressing my concerns experimentally however, I'm still uncertain about the level of the conceptual leap in terms of appropriateness for Nature Communications. Ultimately, I feel the studies are at least soundly executed enough to warrant consideration.

Reviewers' comments:

Reviewer #1 (Remarks to the Author):

I am now happy with the revisions and don't have any further comments to the manuscript.

Reviewer #2 (Remarks to the Author):

The authors addressed the comments of my previous review, however there is still a detail that does not fully convince me.

In reply to my comment on discrepancies in gene expression between SVCs differentiated to adipocytes in Fig. 2J and 2K vs results in supplementary Figure 3d they stated that "The SVCs in Fig 2j and 2k are comparable to cold-exposure conditions Fig 2h and Supplementary Fig 3d because these cells were stimulated with thermogenic stimuli". However, after checking the Experimental section at pg. 18 and 19 (which then mentions the AIM at pg 17 last line of the page) the differentiation conditions of adipocytes do not seem to contain any thermogenic stimulus. If there is no thermogenic stimulus how could they justify the difference between results in Figure 2J and K (experiment with cells aplodeficient in TonEBP) and the results in supplementary Figure 3d (experiment in TonEBP aplodeficient mice at room temperature, i.e. black bars vs. withe bars)? Is there something missing the Experimental section?

Reviewers' comments:

Reviewer #2 (Remarks to the Author):

The authors addressed the comments of my previous review, however there is still a detail that does not fully convince me.

In reply to my comment on discrepancies in gene expression between SVCs differentiated to adipocytes in Fig. 2J and 2K vs results in supplementary Figure 3d they stated that "The SVCs in Fig 2j and 2k are comparable to cold-exposure conditions Fig 2h and Supplementary Fig 3d because these cells were stimulated with thermogenic stimuli". However, after checking the Experimental section at pg. 18 and 19 (which then mentions the AIM at pg 17 last line of the page) the differentiation conditions of adipocytes do not seem to contain any thermogenic stimulus. If there is no thermogenic stimulus how could they justify the difference between results in Figure 2J and K (experiment with cells aplodeficient in TonEBP) and the results in supplementary Figure 3d (experiment in TonEBP aplodeficient mice at room temperature, i.e. black bars vs. white bars)? Is there something missing in the Experimental section?

We apologize for our oversight that we did not provide adequate description of method for beige adipogenesis differentiation of 3T3-L1 and stromal vascular cells. These cells were differentiated into beige adipocytes using indomethacin and triiodothyronine (49).

We added descriptions with new reference (49) as follows: "For induction of beige adipocyte differentiation, beige adipogenesis inducing medium (BAIM) including 1 μ M dexamethasone, 0.5 mM isobutylmethylxanthine, 1 μ M insulin, 125 μ M indomethacin and 1 nM triiodothyronine was used⁴⁹." (p 17, lines 23-25); "SVCs were differentiated to beige adipocytes using BAIM." (pg 18, line 2)

We also revised the legend of Fig. 2 j and k as follows:

"Thermogenic gene (j) and beige marker (k) mRNA abundance in beige adipocytes differentiated from the stromal vascular cells of iWAT (n = 4)."

Reviewer #2 (Remarks to the Author):

Authors provided the explanation to my request regarding data in figure 2j and k as compared to data in figure 2h and supplementary figure 3d and added more details in the experimental section explaining that cells from SVF were differentiated in the presence of thermogenic stimuli (i.e., BAIM including 1 μ M dexamethasone, 0.5 mM isobutylmethylxanthine, 1 μ M insulin, 125 μ M indomethacin and 1 nM triiodothyronine).